



# HOTSSea v1: a NEMO-based physical Hindcast of the Salish Sea (1980 – 2018) supporting ecosystem model development

Greig L. Oldford[1,2], Tereza Jarníková [3], Villy Christensen[1], Michael Dunphy[4]

[1] Institute for Oceans and Fisheries, University of British Columbia, Vancouver, V6T 1Z4, Canada

5  [2] Ecosystem Sciences Division, Fisheries and Oceans Canada, Nanaimo, V9T 6N7, Canada

[3] Tyndall Centre for Climate Change Research, School of Environmental Sciences, University of East Anglia, Norwich, UK

[4] Institute for Oceans Science, Ocean Sciences Division, Fisheries and Oceans Canada, Sidney, V8L 5T5, Canada

10  *Correspondence to*: Greig L. Oldford (greig.oldford@dfo-mpo.gc.ca)





## Abstract

Decadal-scale oceanographic, environmental, and ecological changes have been reported in the Salish Sea, an ecologically productive and biodiverse inland sea in the northeast Pacific that supports the economies and cultures of millions of people; however, there are substantial observational gaps pertaining to physical water properties that make linkages between physical drivers and ecosystem effects difficult to ascertain. With the aim of addressing these gaps, we present the Hindcast of the Salish Sea (HOTSSea) v1 with temporal coverage from 1980 – 2018, developed using the NEMO ocean engine. An inter-model comparison and preliminary evaluation was performed to assess sensitivity to different atmospheric and ocean reanalysis products used for boundary forcings. Biases inherited from forcings were quantified and the effectiveness of a simple temperature bias correction factor applied at one ocean boundary was evaluated. Evaluation of salinity and temperature indicates performance is best in the Strait of Georgia where the model simulates temperature anomalies and a secular warming trend over the entire water column in general agreement with observations. Analyses of modelled ocean temperature trends throughout the northern and central part of the domain where model skill was high and where observations are relatively sparse yielded fresh insights, including that ocean temperature trends are spatially and temporally variable. HOTSSea v1 will support development of an end-to-end spatial-temporal ecosystem model for the Strait of Georgia and has potential for other research and management applications related to decadal-scale climate effects on marine ecosystems, fish, and fisheries.

## Non-Technical Summary

We developed a physical ocean model called the Hindcast of the Salish Sea (HOTSSea) that recreates conditions throughout the Salish Sea from 1980 to 2018, filling in the gaps in patchy measurements. The model predicts physical ocean properties with sufficient accuracy to be useful for a variety of applications. The model corroborates observed ocean temperature trends and was used to examine areas with few observations. Results indicate that some seasons and areas are warming faster than others.

## Copyright





## 1. Introduction

The Salish Sea is an inland sea in the northeast Pacific spanning Canadian and American waters with estuarine characteristics, fjords, and high biodiversity (Harrison et al., 1983; Pata et al., 2022). The productive waters of the Salish Sea support the economy and cultures of a rapidly growing coastal population of 8 – 10 million people including the port cities of Vancouver, British Columbia

(BC, Canada), and Seattle, Washington (United States of America), and dozens of recreational, commercial, and indigenous fisheries (Georgia Strait Alliance, 2020). Many long-term changes of concern associated with regional-scale oceanographic and atmospheric climate processes and global climate change have been reported, including: changes in seasonal wind patterns (Collins et al., 2009; Masson & Cummins, 2007; Preikshot, 2007; Tuller, 2004), precipitation (Beamish, 1993;

Morrison et al., 2002; Yin et al., 1997), ocean water temperatures (Beamish et al., 2010; Masson & Cummins, 2007), properties related to ocean acidification (Jarníková et al., 2022), and river discharge and temperatures (Islam et al., 2019; Martins et al., 2011; Riche et al., 2014). Increasing seasonal stratification and warmer surface waters may also have increased the frequency and duration of harmful algal blooms (Esenkulova et al., 2021; Moore et al., 2015). Changes to regional climate

patterns appear to have increased the variability of the date of the spring phytoplankton bloom (Allen & Wolfe, 2013) which may have led to spatial-temporal mismatches between predators and prey (Allen & Wolfe, 2013; Suchy et al., 2022) and changed the composition of larval fish assemblages (Guan, 2015). Changing ocean conditions are hypothesised to have specifically affected Pacific salmon prey abundance, composition, and spatial-temporal availability via various pathways of effects

(Pearsall et al., 2021) with even small and gradual ocean temperature changes hypothesised to substantially affect the growth, body mass, and marine distributions of fish, generally (Pauly, 2021; Pauly & Cheung, 2018). Several correlative studies link sea surface temperature and stratification with declining survival of several salmon species in the Salish Sea, particularly juvenile coho salmon (*Oncorhynchus kisutch*), Chinook salmon (*O. tshawytscha*), and steelhead (*O. mykiss*; Beamish,

1995; Pearsall et al., 2021; Perry, 2021; Sharma et al., 2013; Sobocinski et al., 2020, 2021; Walters & Christensen, 2019), though correlational analyses are limited by sparse observations and time series that typically pertain to surface waters only.

The patchy nature of oceanic data, particularly as we traverse deeper into historical records, leads to uncertainty about the pace and spatial-temporal patterns of change in the Salish Sea. Physical

hindcasting in this context is a pivotal tool, offering a retrospective lens through which past oceanic conditions are reconstructed. Physical ocean models coupled or linked to biogeochemical and ecosystem models can be used to address data gaps and to explore and evaluate the pathways of effects from water properties to marine ecosystems (Macias et al., 2014; Piroddi et al., 2021). Several ocean and biogeochemical models have been developed for the Salish Sea. These models are either



coarse in resolution for use in the Salish Sea due to a focus on the wider BC coast (Peña et al.,
2016), are focused on Puget Sound (Khangaonkar et al., 2012, 2019; MacCready et al., 2021; Moore
et al., 2015), or incur high computational cost due to high resolution and a focus on shorter term
simulations (Jarníková et al., 2022; Olson et al., 2020; Soontiens et al., 2016; Soontiens & Allen,
2017) making them less well suited for the present purpose of a long hindcast. A lack of products to

use as atmospheric and oceanic forcings has been a further hinderance to developing long hindcasts
for the area. Several ocean and atmospheric reanalyses are now available that are potentially
adequate to facilitate regional hindcasts extending back to 1980 or earlier. However, these products
may lack the spatial-temporal resolution needed for application as forcing for the Salish Sea and
experimental evaluation is therefore needed to check their utility and identify any important biases.

Here, we present HOTSSea v1, developed using the Nucleus for European Modelling of the Ocean
(NEMO) Ocean engine (Madec et al., 2017). We describe and give rationale for the model setup with
attention to three aspects of ongoing model development with particular importance for developing a
long hindcast for the domain: (1) the biases inherited by using various atmospheric and ocean
reanalysis products as surface and boundary forcing, (2) the effect of applying temperature bias

corrections to the open ocean boundary forcing, and (3) a preliminary assessment of model
performance relevant to the aforementioned research applications, including decadal-scale trends.
Priority areas for improvement and further evaluation are also highlighted and, finally, we use the
model to provide a first look at decadal-scale trends in the central and northern portion of the domain
where historical observations are especially sparse.

## 2. Model Overview

    The NEMO ocean engine, version 3.6, supports simulations of ocean dynamics and thermodynamic
processes in three dimensions (Madec et al., 2017). The physical model framework is governed by
primitive equations under hydrostatic balance using the Boussinesq approximation where density
variations are neglected except in their contribution to the buoyancy force (Bourdallé-Badie et al.,

2019). HOTSSea v1 was implemented in a high-performance computing cluster (*Digital Research
Alliance of Canada*, 2022) and the model's scope is limited to physics and hydrodynamics (e.g., tides,
salinity, temperature) - biogeochemistry is not included in HOTSSea v1 due to computational cost.
We used state variables of Practical Salinity (PSU) and Potential Temperature (℃), and the EOS-80
equation of state (Millero, 2010). Sea ice is not included in HOTSSea v1 given that it typically occurs

only in deep inland waters of fjords such as Jervis Inlet. To address issues of omitting ice, we applied
NEMO's ice-if option, where the water temperatures are limited to the local salinity-dependent
freezing point.



For tuning, evaluation, and analysis, a suite of other software and tools were used, including the command line tools *nco* (Zender, 2014) and *cdo* (Schulzweida, 2022); Python 3

(https://anaconda.com) with *pandas*, *SciPy*, *NumPy*, *Cartopy*, and *GSW-Python* packages (Harris et al., 2020; Virtanen et al., 2020); and a custom analysis package written in the Python scripting language used for tuning and evaluation co-developed by one of the co-authors. Other code used to run analyses and produce figures is archived at https://doi.org/10.5281/zenodo.10846149 (Oldford, 2024).




## 2.1. Spatial-Temporal Configuration

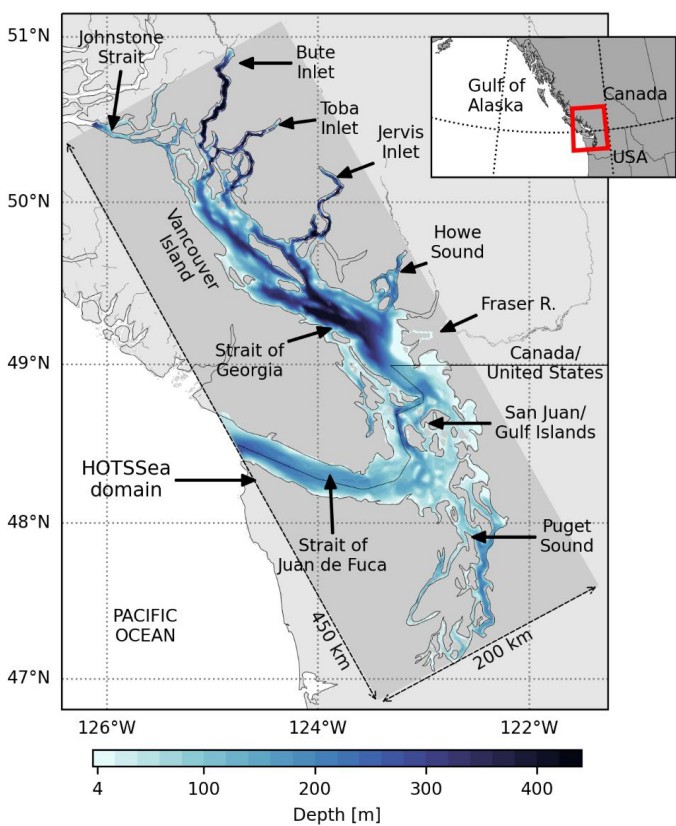

**Figure 1:** Map of model domain showing geographic features, extents of the HOTSSea NEMO model domain (medium grey), and bathymetry.

The model domain of HOTSSea v1 includes several distinct geographic areas within the Salish Sea: the Juan de Fuca Strait, Strait of Georgia, Gulf Islands, and Puget Sound (Figure 1). A key application of the model will be to provide forcings for bioegeochemical and ecosystem models developed to investigate decadal-scale change. The HOTSSea v1 spatial domain was chosen such that it fully encompasses the domain of an ecosystem model under parallel development which

focuses on the Strait of Georgia using the Ecospace model framework (de Mutsert et al., 2023; Walters et al., 1999). The horizontal grid used in NEMO is discretised on a curvilinear orthogonal Arakawa C-grid generalised to three dimensions (Arakawa & Lamb, 1977; Madec et al., 2017). The basic spatial-temporal configuration of HOTSSea v1 began with a previous configuration, SalishSeaCast, implemented at approximately 500 m horizontal resolution for the same domain

(Olson et al., 2020; Soontiens et al., 2016; Soontiens & Allen, 2017). The ~500 m horizontal resolution grid and bathymetry used in the SalishSeaCast model was reduced by a factor of three in





each horizontal direction, taking the mean depth of the neighbouring cells to assign the new depths. The new grid is approximately 1.5 km in horizontal resolution and has a width of ~200 km and length of ~450 km (132 cells x 299 cells; Figure 1). The grid is rotated 29° counter clockwise to true north to

align with the axis of the Strait of Georgia. The bathymetry was processed for SalishSeaCast to avoid sudden changes in depths across grid cells and maintain open channels in narrow passages. We made additional manual edits to maintain channels between islands, maintain connectivity of the main Fraser River channel to the outflow, and avoid erroneously isolating bodies of water. Some narrow water bodies such as Sechelt Inlet, Salmon Inlet, Burrard Inlet, and the Indian Arm fjord are not

resolved in this setup (outlines of these areas are visible in Figure 1). The depths of edited channel cells were approximated from depth averages taken from ~80 m resolution bathymetric data (Pacific Salmon Foundation, 2022). To ensure tidally-driven dynamics were not lost, the main channel of the Fraser River was extended inland by manually adding non-existent river channel cells approximately 150 km in total length, following Soontiens & Allen (2017).

The vertical grid for HOTSSea v1 is divided into 40 vertical ($z$) levels that are gradually stretched to achieve higher resolution at the surface, ranging from 1 m vertical resolution in the upper 10 m to approximately 27 m widths at the deepest level (420 m). Partial steps were enabled to limit large changes in bathymetry between adjacent grid cells. The thickness of each layer is proportionally scaled at each time-step as sea surface height changes using a nonlinear free surface scheme

referred to as the 'variable volume option' (Levier et al., 2007). HOTSSea v1 uses a non-linear free surface option to time-split the solving of the barotropic and baroclinic free surface. The barotropic and baroclinic time steps are set to 6 and 120 s, respectively, and the vertical momentum and tracer advection time stepping set to 2 s. The model was run from 1979-01-01 to 2019-01-01, where the 1980 atmospheric forcings were duplicated and applied to 1979, such that we treat 1979 as a model

spin-up year and exclude it from evaluation. A one-year spin-up was based on a minimum estimate of deep water residency time which elsewhere has been reported to range between one and three years (Pawlowicz et al., 2019). Initial conditions for January, 1979, for temperature and salinity across the domain were generated using climatologies for December and January using SalishSeaCast outputs from 2007 to 2020. An experimental bias correction to the ORAS5 temperature fields was applied

when running the final hindcast.





## 2.2. Boundary Conditions and Forcings

**Table 1:** External forcing used in the model.

| Forcing Dataset | Forcing Type | Model Runs | Temporal extent and resolution | Horizontal Resolution | Citation |
|---|---|---|---|---|---|
| Regional Deterministic Reforecast System (RDRS v2.1) | Surface / Atmospheric | Final | 1980 – 2018; hourly | 0.09°; ~10 km) | Gasset et al., 2021 |
| European Centre for Medium-Range Weather Forecast (ECMWF) Ocean and Sea Ice Re-analysis v5 (ORAS5) | Open ocean boundary conditions | Final | 1975 – 2018; monthly | 0.25°; ~18 km | Tietsche et al., 2017; Zuo et al., 2019 |
| Runoff / River Climatology and Gauge Data | Runoff | Final | 1979 – 2018; hourly and daily | n/a | Morrison et al., 2012; Soontiens et al., 2016 |
| Tidal Constituents | Tidal forcing at open boundaries | Final | n/a | n/a | Soontiens et al., 2016 |
| Coastal Ice Ocean Prediction System (CIOPS) West | Open ocean boundary conditions | Evaluation | 2007 – 2019; hourly | 1/36°; ~2.5 km | Paquin et al., 2020 |
| ECMWF ERA v5 (ERA5) | Surface / Atmospheric | Evaluation | 1979 – present; hourly | 0.28° ; ~31 km | Dee et al., 2011; Hersbach et al., 2020 |
| High Resolution Deterministic Prediction System (HRDPS) | Surface / Atmospheric | Evaluation | 2014 – 2020; hourly | 0.0225°; ~2.5 km | Environment and Climate Change Canada, 2020 |

### 2.2.1. Atmospheric

The Regional Deterministic Reforecast System (RDRS v2.1; Gasset et al., 2021) supplied the atmospheric conditions for forcing the full HOTSSea v1 hindcast. RDRS v2.1 is currently the highest resolution atmospheric reanalysis product available extending back to 1980 (0.09°; ~10 km horizontal).  Two additional atmospheric forcings (Table 1) were evaluated as part of an experimental design: the European Centre for Medium-Range Weather Forecasts (ECMWF) ERA5, a global reanalysis product extending back to 1979 (hourly, at approximately 31 km horizontal resolution; Dee et al., 2011; Hersbach et al., 2020) and the High Resolution Deterministic Prediction System (HRDPS), with spatial coverage of the northern part of North America (Canada and northern United States) with hourly coverage at ~2.5 km horizontal resolution for 2014 - 2020 (Environment and Climate Change Canada, 2020). The RDRS v2.1 product occupies an intermediate horizontal resolution between ERA5 and HRDPS, and the three together offered an opportunity to explore the effect of horizontal resolution of atmospheric forcing on model performance in the Salish Sea.





### 2.2.2. Open Boundaries

There are two boundaries that connect the Salish Sea to the Pacific Ocean: the mouth of Juan de
Fuca Strait in the southwest and Johnstone Strait in the north (Figure 1). To first evaluate the effects
of using different ocean boundary forcings at the mouth of the Juan de Fuca Strait, a higher resolution
model, CIOPS-West (Paquin et al., 2020), was used in shorter evaluation runs (horizontal resolution 2
- 2.5 km; 1/36°; Table 1Table 2). The Ocean Reanalysis System 5 (ORAS5; Tietsche et al., 2017;
Zuo et al., 2019) was the only available reanalysis product with coverage for the full model hindcast
and was used to supply ocean open boundary conditions in the final model. ORAS5 has a horizontal
resolution at the latitude of the Salish Sea of approximately 18 km (0.25°). At the northern boundary
(Johnstone Strait), we used a monthly climatology of temperature and salinity (Dosser et al., 2020,
2021).

### 2.2.3. River Discharge and Runoff

River input into the Salish Sea periodically creates a brackish layer extending across the Strait of
Georgia and drives strong estuarine circulation via Juan de Fuca Strait (Harrison et al., 1983). The
Fraser River is the largest single source of freshwater influx into the domain and supplies
approximately two thirds of the total annual freshwater input (Pawlowicz et al., 2019). Fraser River
discharge is monitored as part of a long-term program (Morrison et al., 2012). Following Soontiens &
Allen (2017), we used available flow records for the Fraser River from gauges approximately 150 km
inland at the city of Hope, BC (Water Survey of Canada, 2015), and supplemented the Fraser River
flow data with climatological data for additional freshwater input downstream of the station. A
climatology was used for Fraser River runoff temperatures (Morrison et al., 2002) due to a lack of
long-term measurements from the lower Fraser. The location of river outflow for the Fraser River was
placed in the main channel before the river branches into a delta (at the town of Delta, BC). All other
river outflows were assigned to the grid cell closest to the river mouth. Many rivers other than the
Fraser are not monitored, so climatological patterns for discharge and temperature for 150 rivers
flowing into the Salish Sea were used (Morrison et al., 2012). We adapted the input file containing
these river input data from the ~500 m horizontal resolution model grid used by Soontiens et al.
(2016) to the ~1.5 km horizontal resolution used here and adjusted the outflow locations as required.

### 2.2.4. Tides

At the two open boundaries, tides were forced with eight tidal constituents (K1, O1, P1, Q1, M2, K2,
N2, and S2). Tidal heights and currents at the Juan de Fuca boundary were originally taken from
WebTide (Foreman et al., 2000) and then manually tuned (Soontiens et al., 2016). At the northern





open boundary sea surface height and tidal harmonics were forced for the major M2 and K1
constituents and SSH harmonics for the O1 and S2 harmonics were configured using calculations
from Thomson & Huggett (1980) with remaining constituents taken from WebTide and subsequently
tuned.

## 3. Model Evaluation

Observations were collated from various instruments and sources (Table 2) and used to do a
preliminary evaluation of the model's performance with respect to sea surface temperature (SST),
sea surface salinity (SSS) and temperature and salinity over depths. To understand the trade-offs
between spatial-temporal resolution, tractability, and model skill we used an experimental approach
where forcings were incrementally swapped to help with isolating the most likely source of model

error and bias. Version 201905 of the SalishSeaCast model was used as a baseline for evaluating the
effect on overall model performance of changing the spatial-temporal setup. In the final HOTSSea v1
model we evaluated the modelled long-term temperature trend against observations at Nanoose
station, the only long-term dataset with at least biweekly depth profiles done in the model domain
extending back to the beginning of the hindcast (Table 2).




**Table 2:** Summary of data used for model evaluation.

| Instrument Type | Dataset Title | Variables | Observations (N) | Description | Source |
|---|---|---|---|---|---|
| **Conductivity, Temperature, and Depths (CTD) Casts** | Fisheries and Oceans Canada's (DFO) Institute for Ocean Science (IOS) CTD casts dataset | Conductivity, Temperature, Depth, Pressure, Oxygen and Salinity | 24,810 | Contains CTD measurements collected in the Central Strait of Georgia, British Columbia, Canada using rosette mounted CTDs. | DFO, 2022c |
| | DFO IOS | Salinity, Temperature, Depth, Pressure | 3,942 | Surveys conducted from 1965 to present and include Nanoose Bay station, a Canadian military CTD dataset which were provided upon request from DFO. | Personal communication (M. Dunphy) and WaterProperties.ca |
| | Hakai Institute | Salinity, Temperature, Depth, Pressure | 2,871 | CTD data collected from 2012 to present by the Hakai Institute in waters surrounding Calvert Island, Johnstone Strait, and Quadra Island areas. | Jackson et al., 2021 |
| | Pacific Salmon Foundation (PSF) | Salinity, Temperature, Depth, Pressure | 3,437 | CTD casts collected by PSF for Strait of Georgia. | Pacific Salmon Foundation, 2023 |
| **Lightstation (LS) Near-Surface Water Properties** | Environment and Climate Change Canada (ECCC) | Temperature, Salinity | 7 | Observations from lightstations where daily sea-surface temperature and salinity measurements have been collected from 1914 to present. Measurements were made daily using seawater collected in a bucket lowered into the surface water at or near the daytime high tide. | DFO, 2022a; Treasury Board Secretariat, 2023 |
| **Wave Buoys** | ECCC via DFO | Sea Surface Temperature (SST) | 5 | Wave and temperature data from buoys. Sea surface temperature data have undergone automated quality control. Historical data are merged with real-time acquisition. | Fisheries and Oceans Canada (DFO), 2024 |



## 235  3.1. Experimental Evaluation

The years chosen for running preliminary experimental evaluations were 2016 – 2018. The available forcing data and models had coverage for those years and generally a larger volume of evaluation data is available for the most recent years (Table 3). Experimental runs of the HOTSSea model were given run codes. The first was HOTSSea v0.1, which used the highest resolution atmospheric and

ocean boundary forcings available. This run was used to do a comparative evaluation with SalishSeaCast v201905 which has a higher horizontal resolution (~500 m versus ~1500 m). The LiveOcean model (Fatland et al., 2016) forcings used in SalishSeaCast at the JFS open ocean boundary were not available for the 2016 – 2018 period so in HOTSSea v0.1 we used CIOPS-W BC12, a model also developed using the NEMO v3.6 ocean engine covering the northeast Pacific at

an approximate horizontal resolution of 2.0 - 2.5 km (Paquin et al., 2020). The HOTSSea v0.12 run was used to evaluate the effect of swapping from HRDPS to the ERA5 atmospheric forcings (~31 km horizontal; Dee et al., 2011; Hersbach et al., 2020). At the time HOTSSea development began in 2021, ERA5 was the only climate reanalysis product available for the entirety of the hindcast period. At the time of writing, it is still the only reanalysis extending back to the 1940s and therefore

evaluation of this product for use for atmospheric forcing was a priority. The HOTSSea v0.14 and HOTSSea v0.16 experiments helped evaluate the effect of using the ORAS5 (~18 km horizontal; Tietsche et al., 2017; Zuo et al., 2019) dataset for ocean boundary conditions at the mouth of Juan de Fuca Strait. The HOTSSea v0.18 experiment used the RDRS v2.1 atmospheric outputs for forcing, which have an intermediate horizontal resolution of ~10 km (Gasset et al., 2021). To evaluate each

experiment, we used data, methods, and statistics as described in the next section. Model performance was evaluated using results aggregated over the 2016 - 2018 period - analyses were also carried out on model results grouped by month and year, though only results aggregated for the entire period are presented here and only the results using CTD measurements are highlighted here for brevity.






**Table 3:** Experimental evaluation run codes and forcings.

| Model | Version or Run Code | Evaluation Purpose | Years | Surface Forcing | Ocean Boundary Forcing | Reference |
|---|---|---|---|---|---|---|
| SalishSeaCast | v201905 | Comparison of model performance at higher horizontal resolution | 2016 - 2018 | HRDPS | LiveOcean at JFS boundary, climatology at northern boundary | Soontiens et al., 2016; Soontiens & Allen, 2017; Olson et al., 2020 |
| Coastal Ice-Ocean Prediction System for the West Coast of Canada (CIOPS-W) | BC12 | Comparison of model performance at lower horizontal resolution | 2016 - 2018 | HRDPS1 (2.5km) / RDPS2 (10km) | Regional Ice Ocean Prediction System (RIOPS) v2 | Paquin et al., 2020 |
| HOTSSea | v0.1 | Comparison with two models listed above using ~1.5 km² horizontal resolution | 2016 - 2018 | HRDPS | CIOPS-West | This study |
| HOTSSea | v0.12 | Evaluate sensitivity to lower resolution atmospheric forcing | 2016 - 2018 | ERA5 | CIOPS-West | This study |
| HOTSSea | v0.14 | Evaluate sensitivity to lower resolution ocean boundary and atmospheric forcings | 2016 - 2018 | ERA5 | ORAS5 | This study |
| HOTSSea | v0.16 | Evaluate sensitivity to lower resolution ocean boundary forcings | 2016 - 2018 | HRDPS | ORAS5 | This study |
| HOTSSea | v0.18 | Evaluate sensitivity to intermediate resolution atmospheric and lower resolution boundary forcings | 2016 - 2018 | RDRS v2 | ORAS5 | This study |
| HOTSSea | v1.01 | Final run (full hindcast period) | 1979 - 2018 | RDRS v2.1 | ORAS5 | This study |
| HOTSSea | v1.02 | Evaluation of boundary condition correction (full hindcast period) | 1979 - 2018 | RDRS v2.1 | ORAS5 (temp bias adjusted) | This study |



## 3.2. Model-Observations Evaluation Methods

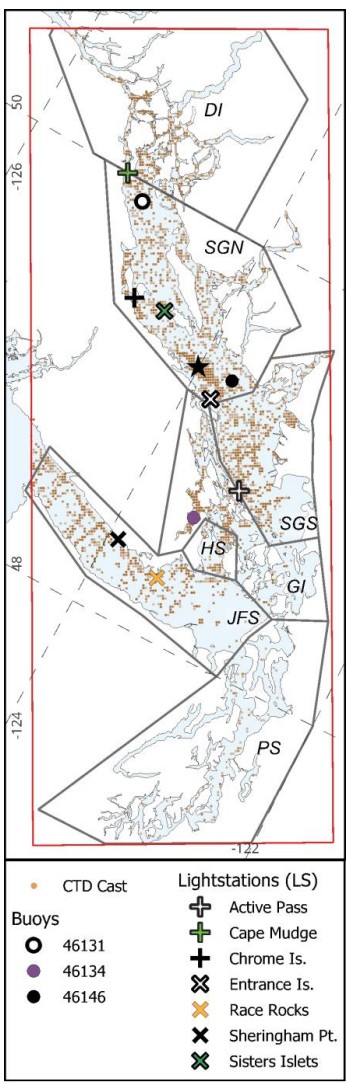

**Figure 2:** Map of HOTSSea v1 model domain (red rectangle) and subdomains used for analysis (grey polygons; DI = Discovery Islands, SGN = Strait of Georgia North, SGS = Strait of Georgia South, HS = Haro Strait, GI = Gulf Islands, JFS = Juan de Fuca Strait, PS = Puget Sound). Locations of CTD casts are indicated by orange stippling (darker denotes higher density), and Nanoose station is indicated by the black star.

The final hindcast was evaluated using the datasets grouped by subdomains (Table 2, Figure 2). Subdomains were selected based on distinct geographic features, data availability, and physical characteristics.





### 3.2.1. Vertical Profiles

CTD casts were acquired from various sources (Table 2). After quality control, 27,272 CTD casts were used in the analysis of the final hindcast. These data have heterogeneous spatial coverage

when aggregated by subdomain (Figure 2; Table 4). For the model intercomparison and experimental evaluation only CTD data from 2016 - 2018 were used ($N = 8,012$), which also had spatially heterogenous coverage across subdomain.

**Table 4:** Spatial distribution of CTD data used for evaluation of entire hindcast (1980 - 2018) and shorter experiments (2016 - 2018).

|  | CTD count | |
| --- | --- | --- |
| *Subdomain* | **1980 - 2018** | **2016 – 2018** |
| *Discovery Islands (DI)* | 3,649 | 1,884 |
| *Strait of Georgia North (SGN)* | 12,365 | 3,080 |
| *Strait of Georgia South (SGS)* | 4,140 | 1,517 |
| *Gulf Islands (GI)* | 2,512 | 871 |
| *Hare Strait (HS)* | 800 | 184 |
| *Puget Sound (PS)* | 99 | 23 |
| *Juan de Fuca Strait (JFS)* | 3,707 | 453 |
| *Total :* | **27,272** | **8,012** |

The closest model grid cell and time index was found for each CTD measurement and the measurements over depths for each CTD cast were vertically interpolated to the model depth levels. The model error was then calculated for each model-observation pair (*m, o*) of time series (*Error = m*

*– o*). The bias is the mean error, given by:

$$\text{Bias} \ = \ \overline{m} - \overline{o} \tag{1}$$

where the mean of the observations and model are denoted by $\bar{o}$ and $\bar{m}$ respectively. The RMSE is given by:

$$RMSE \ = \ \sqrt{\frac{1}{N}\sum_{i=1}^{N}(m_i - o_i)^2} \tag{2}$$


where the set of measurements across all casts for a given depth stratum, time frame, and subdomain denoted by *N*. The bias is often used to infer the accuracy of a model whereas RMSE helps assess the precision of a model. The CRMSE is given by:

$$\text{CRMSE} \ = \ \sqrt{\frac{1}{N}\sum_{i=1}^{N}\big((m_i - \overline{m}) \ - \ (o_i - \overline{o})\big)^2} \tag{3}$$



where $o_i$ denotes a single observation out of a set of observations, $N$, with the corresponding model output denoted by $m_i$. The CRMSE quantifies the variability of the model as compared with observations. The WSS statistic is a dimensionless measure of model skill, ranging from zero denoting poor agreement between model and observations and one denoting perfect agreement (Willmott, 1981):


$$WSS = 1 - \frac{\sum_{i=1}^{N}(m_i - o_i)^2}{\sum_{i=1}^{N}(|m_i - \overline{o}| + |o_i - \overline{o}|)^2} \qquad (4)$$

To calculate the statistics above, the CTD data were first grouped by subdomain, period, and depth strata during analysis. We highlight below only the results of grouping the data first by subdomain and model depth level and second by subdomain and selected depth strata (0 -> 30 m; 30 -> 150 m; >

150 m; all depths). For statistics grouped by depth strata, the depth-integrated mean from individual CTD casts within each depth grouping were first calculated. These values were treated as a single measurement ($o_i$) in the set of CTD casts, $N$, across each subdomain with $\overline{o}$ representing the mean of depth-integrated means. Model results were extracted for each observation and depth-integrated in the same manner.

## 3.2.2. Sea Surface Temperature and Salinity

Sea surface temperature (SST) and salinity (SSS) were evaluated using measurements collected at high tide during daylight hours by lighthouse staff at seven lighthouses throughout the domain (Figure 2; Treasury Board Secretariat, 2023). Lightstation data (LS) were available for the entirety of the hindcast period for Chrome Island, Entrance Island, and Race Rocks lighthouses whereas others had

partial coverage (

Table *5*). The time of day when samples were taken was not always provided in the dataset. As such, the closest tidal gauge each day was found for each lighthouse and the high tide time was extracted and then matched to the lighthouse sample data as required.


**Table 5:** Lightstation data summary used for evaluation of SST and SSS.

| Lightstation ID | Years | Location (subdomain) |
|---|---|---|
| Active Pass | 1980 – 2011 | SGS |
| Cape Mudge | 1980 – 1985 | SGN |
| Chrome Island | 1980 – 2019 | SGN |
| Entrance Island | 1980 – 2018 | SGN |
| Race Rocks | 1980 – 2018 | JFS |
| Sheringham Point | 1980 – 1988 | JFS |
| Sisters Islets | 1980 - 2008 | SGN |





Sea surface temperature measurements were also available from buoys within the model domain (Figure 2). Canadian buoy data were downloaded from online repositories (Fisheries and Oceans

Canada (DFO), 2024; NOAA National Buoy Data Centre, 2023). The buoy data we prepared had heterogeneous temporal coverage of the hindcast period (Table 6). For the evaluation of the full hindcast we only used buoys with ten or more years of data (buoy IDs: 46146, 46131, 46134).

**Table 6:** Buoy data summary used for evaluation of SST.

| Buoy ID | Years | Location (subdomain) |
|---------|-------------|---------------------|
| 46131 | 1992 - 2019 | SGN |
| 46134 | 2001 - 2019 | GI |
| 46146 | 1992 - 2019 | SGN |


The statistics used for evaluation of SST and SSS include

$$\sigma_m^2 = \frac{1}{N}\sum_{i=1}^{N}(m_i - \overline{m})^2 \tag{5}$$

$$R = \frac{1}{\sigma_o \sigma_m}\frac{1}{N}\sum_{i=1}^{N}(m_i - \overline{m})(o_i - \overline{o}) \tag{6}$$

where $\sigma$ is the standard deviation of the model (or observations) and $R$ refers to Pearson's $R$ (i.e., correlation coefficient). For plotting of results using Taylor diagrams, the relationship between $\sigma$, $R$,

and *CRMSE* is used:

$$CRMSE = \sqrt{\sigma_o^2 + \sigma_m^2 - 2\sigma_o\sigma_m R} \tag{7}$$

The $\sigma_m$ and CRMSE statistics were normalised by scaling to the standard deviation of the observations to allow display of evaluation results for multiple stations on the same plots:


$$\sigma_m' = \frac{\sigma_m}{\sigma_o} \tag{8}$$

$$NCRMSE = CRMSE / \sigma_o \tag{9}$$

Note that in target plots, the NCRMSE has been modified by the sign of $\sigma_m$ - $\sigma_o$ (Kärnä et al., 2021).

## 3.3. Ocean Boundary Temperature Bias Correction

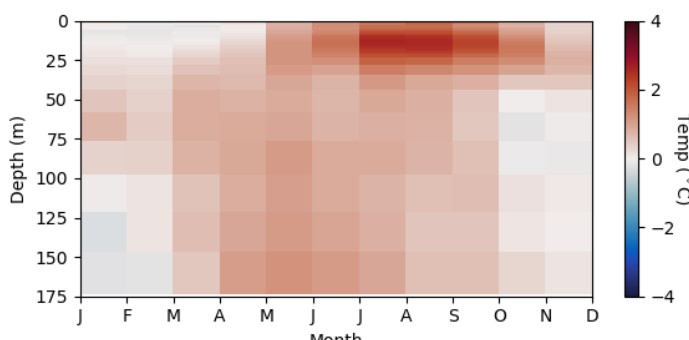


**Figure 3:** Monthly climatology of ORAS5 temperature bias over depths at Juan de Fuca open ocean boundary.

The ORAS5 global ocean reanalysis extends back to 1958 (Copernicus Climate Change Service, 2021) and it would be ideal to extend the model back to 1958, too; however, the magnitude of biases

that would be inherited from ORAS5 was hitherto unknown. Based on the results of experiments described above, we suspected biases could be substantial. To investigate, we collated observations from CTD instruments sampled between 1980 and 2018 from the mouth of the Juan de Fuca Strait ($N$ = 2,162) and compared these observations to temperatures in the ORAS5 data. We chose to do this for the entire hindcast period rather than using the experiments for only 2016 - 2018 given the

observational data available at this location for the experimental period were relatively limited. ORAS5 outputs were interpolated from monthly to daily using the *cdo* toolset (Schulzweida, 2022), which is the same procedure done by NEMO internal routines during model runs. Each CTD measurement was then matched to the closest ORAS5 grid point. Both the observations and the ORAS5 model data were interpolated vertically to the HOTSSea model depth levels. The ORAS5

model bias was calculated using monthly mean bias at each depth level.

The analysis indicated that ORAS5 at the JFS boundary is biased warm most depths and months except January with the mean summer bias near the surface approaching +4 °C (Figure 3). Surface waters in the ORAS5 model were biased fresh in the spring and summer, especially in the top 10 m

where the mean bias approached -4 PSU. As a first step towards a more comprehensive bias correction, the mean monthly temperature bias for each depth level was used as a correction factor applied it to the boundary conditions. We chose to prioritise temperature to isolate a single variable and because applying a bias correction factor to salinity would run a risk of introducing dynamic instability. The temperature bias correction method is acknowledged here to be a crude approach

compared to various alternatives (Adachi & Tomita, 2020). The model run with ocean boundary temperature bias correction factor applied to ORAS5 is referred to as HOTSSea v1.02. We then





evaluated the change in model performance versus with no temperature bias correction (HOTSSea v1.01).

## 3.4. Trend Analysis

The ability of the HOTSSea model to recreate observed long-term ocean temperature trends is one of the most important tests of the utility of the model. The only station in the model domain where measurements over the entire water column were collected with bi-weekly regularity for the entirety of the hindcast is the Nanoose station, located in the central Strait of Georgia (Figure 2). At this location, CTD casts have been sampled approximately every two weeks since 1979 and with less regularity

back to the late 1960s. Trends in water temperature at Nanoose station were previously analysed and estimated to be 0.24 +/- 0.1 ℃ decade$^{-1}$ between 1970 and 2005 (Masson & Cummins, 2007; MC07). We acquired the same Nanoose station data as analysed by MC07 with updates for recent years from a digital archive at DFO Institute of Ocean Sciences (IOS) and in the DFO IOS Water Properties Database in August, 2022. To cross-check our analysis with MC07, we also calculated the slope for

the same period as that study (1970 – 2005) similarly using ordinary least squares linear regression and found a similar result: a warming trend over the water column (4.5 – 400 m) of 0.257 ℃ / decade was assessed here versus 0.24 ℃ / decade in MC07. We attribute the difference to the application of a boxcar filter in MC07 to the anomalies in the previous study.

To compare temperature anomalies and trends observed at Nanoose station with those predicted by

the model, first obviously erroneous measurements (e.g., with coordinates on land or depths exceeding the maximum depth of ~400 m at Nanoose station) were removed and units were converted to match those used in HOTSSea. The number of CTD casts ultimately used was 5,692. Measurements taken in each CTD cast were interpolated to match depth strata of the closest HOTSSea grid cell. The mean for each depth bin grouped in two-week time intervals was calculated

for the hindcast period. A climatology over depth and time was generated for the hindcast period by taking the mean across years for each depth bin and two-week block. Time series of temperature and salinity anomalies were calculated by differencing the depth-binned data and the climatology. A 'blind' approach was then used to match each CTD cast to HOTSSea outputs where model data were only extracted for dates, times, and locations corresponding to each CTD measurement (i.e., gaps in the

Nanoose time series also were present in the extracted model results). The same procedure as above was then used for the model results to generate climatologies and anomalies.

Analysis of temperature trends was conducted using the Nanoose station data and model results drawn from the same area to examine the performance of HOTSSea v1.0x at simulating long-term trends. Depths shallower than 4.5 m were omitted because the first depth at which measurements

were taken was inconsistent (following MC07). The analysis of the depth integrated anomaly trend



over the entire water column (4.5 – 400 m) was first done, followed by analysis of the trend at each model depth. To quantify the magnitude of a linear trend, the Theil-Sen slope was used (TS; Sen, 1968; Theil, 1950). The TS approach is more robust to data gaps, outliers, and non-normal and heteroscedastic residuals than ordinary least squares linear regression (LR; Wilcox, 1998). To

calculate TS, the median slope of all data pairs is found:

$$TS = median \left[ \frac{x_i - x_j}{j - i} \right] \text{ for all i < j} \tag{10}$$

where $(x_i, x_j)$ is a pair of values in the ordered time series ($j > i$). The LR method used in MC07 and the TS method were compared and we estimated the trend for the 1970 – 2005 period using TS to be 0.256 ℃ / decade, effectively the same as the LR method after accounting for measurement precision (see above). The similarity between trends estimated between the LR and TS method is consistent

with findings in a previous analysis of SST in the same region (Amos et al., 2015).

Detrended residuals were analysed for periodicity, autocorrelation, non-normal distribution, and heteroskedasticity prior to choosing the test used for determining statistical significance. The data were first 'deseasoned' using the climatological mean to produce anomalies, as described above. The deseasoned anomalies were detrended by removing the secular trend calculated using the TS

method and fast Fourier transform (FFT) was used to detect any remaining periodicities in the detrended and deseasoned residuals by examining the top five peaks in the power spectrum (Bluestein, 1970; Cooley & Tukey, 1965) using *scipy* (Virtanen et al., 2020) and *numpy* Python packages (Harris et al., 2020). The presence of autocorrelation was evaluated by modelling the residuals as a first order autoregressive process (AR-1) and computing the autocorrelation function

(ACF) using Bartlett's method for computing the 95% confidence interval (Brockwell & Davis, 2016; Parzen, 1964) with the *statsmodels* package in Python (Seabold & Perktold, 2010). The residuals were tested for normality using the Shapiro-Wilk test where *p* value of <=0.9 was interpreted as insufficient evidence to reject a non-normal distribution (Shapiro & Wilk, 1965) using *scipy*. Heteroskedacity was evaluated using White's test (White, 1980) and the Goldfeld-Quandt test

(Goldfeld & Quandt, 1965) using *statsmodels.*

The analysis using FFT revealed the presence of periodicity in residuals of frequencies of 5.1 and 17.7 years, confirming that the data were de-seasoned at the sub-annual scale but suggesting climate modes operating at longer time scales may limit the ability of accurate secular trend estimation, a similar result to previous studies (Amos et al., 2015). The 17.7 periodicity detected is

consistent with the PDO, the NPGO, or a combination of the two. The 5 year period is approximately consistent with the ENSO cycle (Andres Araujo et al., 2013; McPhaden et al., 2006).





Analysis of the data revealed that the residuals of the detrended anomalies (both model and observation) had significant autocorrelation, were not normally distributed, and with some heteroskedasticity detected at some depths. The Mann-Kendall (MK) method (Kendall, 1948; Mann, 1945), a non-parametric method, was chosen for evaluating trend significance. This method is widely used in climatological and meteorological applications (Amos et al., 2015; Gocic & Trajkovic, 2013) and has the advantage of being relatively robust to non-normal data distributions and the data gaps present in the Nanoose time series. To check for statistical significance using MK, first the 'S' statistic is calculated:

$$S = \sum_{i=1}^{n-1} \sum_{j=i+1}^{n} sgn(x_j - x_i) \tag{11}$$

where $n$ is the number of data points in the time series and $sgn(x_j - x_i)$ representing the sign function:

$$sgn(\theta) = \begin{cases} +1, if\ \theta\ >\ 0; \\ 0, if\ \theta\ =\ 0; \\ -1, if\ \theta\ <\ 0 \end{cases} \tag{12}$$

When the $S$ statistic is large it indicates that values earlier in the time series tend to be smaller than those later, and the trend tends positive. Next, the variance of the test statistic is required, which accounts for ties in the data:

$$VAR(S) = \frac{1}{18}\left[ n(n-1)(2n+5) - \sum_{p=1}^{m} t_p(t_p - 1)(2t_p + 5) \right] \tag{13}$$

where $n$ is the length of the time series, $m$ is the number of tied values, and $t_p$ is the number of ties in the $p$th tied value. Using S and VAR(S), one computes the standardised normal variate Z:

$$Z_{MK} = \begin{cases} \dfrac{S-1}{\sqrt{VAR(S)}}, if\ S\ >\ 0; \\ 0, if\ S\ =\ 0; \\ \dfrac{S+1}{\sqrt{VAR(S)}}, if\ S\ <\ 0 \end{cases} \tag{14}$$





The standardised statistic, $Z_{MK}$, follows the normal distribution with $E(Z) = 0$ and $V(Z) = 1$. The null hypothesis – that there is no trend – may be rejected if the absolute value of $Z$ is larger than the theoretical value of $Z_{1-\alpha/2}$ for a one-tailed test or $Z_{1-\alpha}$ for a two-tailed test (used here), where α is a chosen statistical significance level, which here was 95%. The $Z$ values were also used for lower

confidence limits (LCL) and upper confidence limits (UCL). The MK test, similar to LR with t-test, requires the data be independent and therefore serial autocorrelation must be dealt with in advance (Helsel & Hirsch, 1992). Yue et al., (2002) proposed a 'pre-whitening' procedure to remove the effect of serial autocorrelation prior to estimating the trend significance using MK. When autocorrelation was present, we used the '3PW' algorithm (Collaud Coen et al., 2020) which combines three pre-

whitening methods to minimise risks of type I and type II errors which we applied using the *mannkendall* (v1.1.1) Python code.

After comparing the model with the measurements from the Nanoose dataset, we evaluated the trend in temperature and salinity for several depth strata (< 30 m, 30 - 150 m, > 150 m, and all depths) in each grid cell in the subdomain using the same statistical methods described above, though the 3PW

method was not applied due to computational cost. The analysis was limited to the Strait of Georgia and surrounding waters for several reasons: (i) preliminary results indicated that model performance was best in this part of the domain, (ii) observations collated have relatively more coverage in this area, (iii) and it is the focus of an ecosystem model under development in parallel. Mean weekly water temperatures were calculated for each grid cell, season ('winter' = December, January,

February; 'spring' = March, April, May, 'summer' = June, July, August; 'fall' = September, October, November), and for each depth group. When generating plots, grid cells were masked that were shallower than a threshold set for each depth strata (thresholds: 20 m for 0->30 m; 150 m for 30->150 m; 200 m for >150 m) and coloured grey.



# 4. Results

## 4.1. Experimental Evaluation


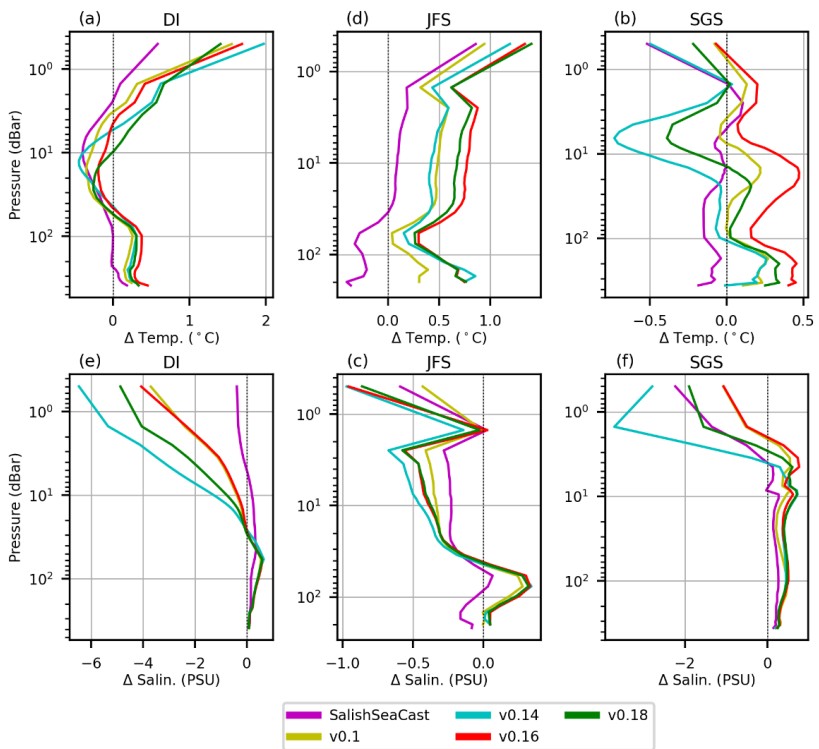

**Figure 4:** Model bias (model - observations) over depths using CTD data for short (2016 - 2018) experimental runs, highlighting temperature (top row) and salinity (second row) bias and CRMSE over depths (shading). HOTSSea v0.18 corresponds to the setup used in the final model run without bias correction (v1.01). See Figure 2 for map of subdomains.

In the first model experiment (years 2016 – 2018), HOTSSea v0.1 performance was compared with the higher resolution SalishSeaCast model. A notable difference was that HOTSSea v0.1 had a mean bias of 0.14 °C over all depths in the SGS subdomain versus -0.017 ˚C in SalishSeaCast (Figure 4). Large differences were noted in the northernmost DI subdomain where near-surface (0 - 1.5 m) biases were larger in HOTSSea v0.1 relative to SalishSeaCast, approaching +2 °C and -4 PSU at 0.5

m. We attribute the warm bias in the DI subdomain to the 3x coarsened HOTSSea model grid relative to SalishSeaCast, limiting the model's ability to resolve the relatively narrow topography in the DI. At depths > 1.5 m HOTSSea v0.1 performed well with mean biases over the water column of -0.12 °C and -0.4 PSU. In the JFS subdomain, HOTSSea v0.1 performed similarly to SalishSeaCast except for the introduction of a temperature bias of 0.3 - 0.5 ˚C at depths > 2 m. The increase in bias was most

pronounced in the JFS subdomain at depths > 20 m but also present in the Strait of Georgia South



(SGS) subdomain. The boundary forcings were hypothesised to be the main culprit, since different boundary forcings were used here (CIOPS-W) than in SalishSeaCast (LiveOcean).

The HOTSea v0.12 run yielded similar results to v0.14 and therefore for brevity results from the latter
were omitted from plots (Figure 4). HOTSSea v0.14 and HOTSSea v0.16 were used to evaluate the impact to model skill of using ORAS5 (~18 km horiz.) for boundary conditions at the Juan de Fuca Strait open ocean boundary versus CIOPS-W (~2.5 km horiz. res.) and the effect of using ERA5 (~31 km horiz. res.) for atmospheric forcing instead of HRDPS (~2.5 km horizontal), respectively. HOTSSea v0.16 outputs had systemic biases with respect to temperature in the JFS subdomain:
approximately +0.5 °C at 10 m, increasing to +0.8 °C at > 100 m. The model's performance with respect to salinity in JFS remained similar to HOTSSea v0.14 (Figure 4). Looking at the SGS subdomain, the warm bias was also apparent at depths greater than 10 m, though less severe than in JFS (approximately +0.3 °C at 10 m depth, increasing to +0.5 °C at 100+ m depths). In the northern-most subdomain, the Discovery Islands (DI) model performance was similar between the HOTSSea
v0.14 and v0.16 experiments with respect to salinity and temperature over depths. Our interpretation is that the model is inheriting a warm bias in mid and deep waters from the ORAS5 boundary conditions, evidenced by decreasing temperature bias with increasing distance from the open ocean boundary. Contrary to expectations, swapping the *lower* resolution ERA5 outputs for the *higher* resolution HRDPS outputs for atmospheric conditions in the HOTSSea v0.14 experiment led to *less*
of a warm temperature bias across all depths in the JFS subdomain at depths up to 100 m while having little discernible effect on salinity. The improved performance in JFS was unexpected because the model domain is relatively poorly resolved by ERA5 versus HRDPS. This result prompted comparisons of the atmospheric products with respect to wind speed where it was found that ERA5 wind speeds are generally weak and less variable relative to the higher resolution HRDPS winds. We
suspect that wind-driven vertical mixing is biased low when using ERA5 and therefore masks the effect of a near-surface warm bias by keeping the biased-warm water closer to the surface. Further investigation is warranted and understanding the sources of biases is a priority.

In the final experimental run, HOTSSea v0.18, the RDRS v2.1 (~10 km horizontal) atmospheric
conditions were used, representing an intermediate between the highest resolution HRDPS forcings available only for 2014 - 2020 and the lower resolution ERA5 forcings available for the entire hindcast period. The run performed similarly or better than the previous two runs with respect to temperature and salinity in most subdomains. The JFS subdomain was an exception where the warm temperature bias was greater than in HOTSSea v0.14. As above, the bias increase was unexpected since
HOTSSea v0.14 used the relatively weak and poorly resolved ERA5 atmospheric conditions. Thus, using both higher resolution and presumably more accurate atmospheric products (RDRS and HRDPS) resulted in *greater* temperature bias over depths up to ~80 m in the JFS subdomain. This





result is consistent with the diagnosis we reached above that the ERA5 atmospheric forcing fields
used in the HOTSSea v0.14 experiment masked the effects of a warm temperature bias in the JFS
subdomain up to depths of ~10 m due to weaker winds and a shallower mixing as compared to the
more powerful HRDPS and RDRS winds in the HOTSSea v0.14 and HOTSSea v0.18 experiments,
respectively.

These preliminary experiments illuminated several important factors affecting model performance
related to atmospheric and ocean boundary forcings. Coarsening the model bathymetry from 500 m
to 1.5 km in the horizontal has especially impacted model performance in areas with narrow
topography but otherwise has had only a minor effect on the evaluated aspects of model
performance. Puget Sound (PS) remains essentially unevaluated due to a lack of collated data for the
area but we suspect a similar issues to the DI subdomain due to narrow topography. An important
takeaway was an apparent systemic temperature bias present in ORAS5, the only reanalysis for
hindcasting the ocean boundary conditions to 1980 and prior available at the time of writing. The
ERA5 product similarly was deemed to be inadequate for the present purposes due to underpowered
winds throughout the domain (RDRS v2.1 was used instead). The experiments prompted us to verify
and quantify the bias introduced at the ocean boundary and to determine if simple bias correction
could hold potential to reduce or eliminate inherited biases (results below).





## 4.2. Bias Correction of Open Ocean Boundary Conditions

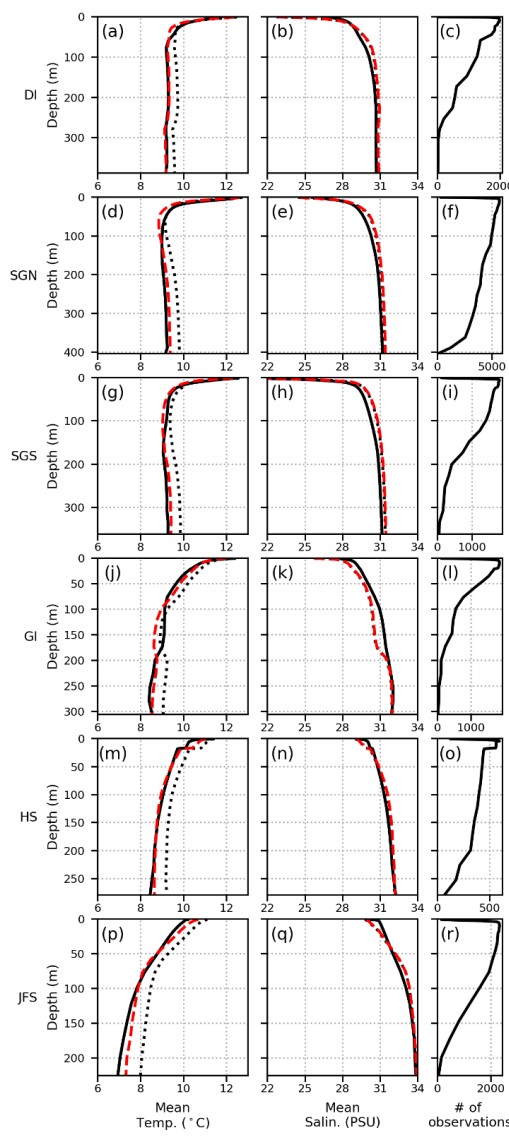

**Figure 5:** Results of analysis using CTD data, aggregated by subdomain, showing mean temperature and salinity over depths (solid line = observations, dashed black line = HOTSSea v1.01 without bias correction, dashed red line = HOTSSea v1.02 with bias correction)




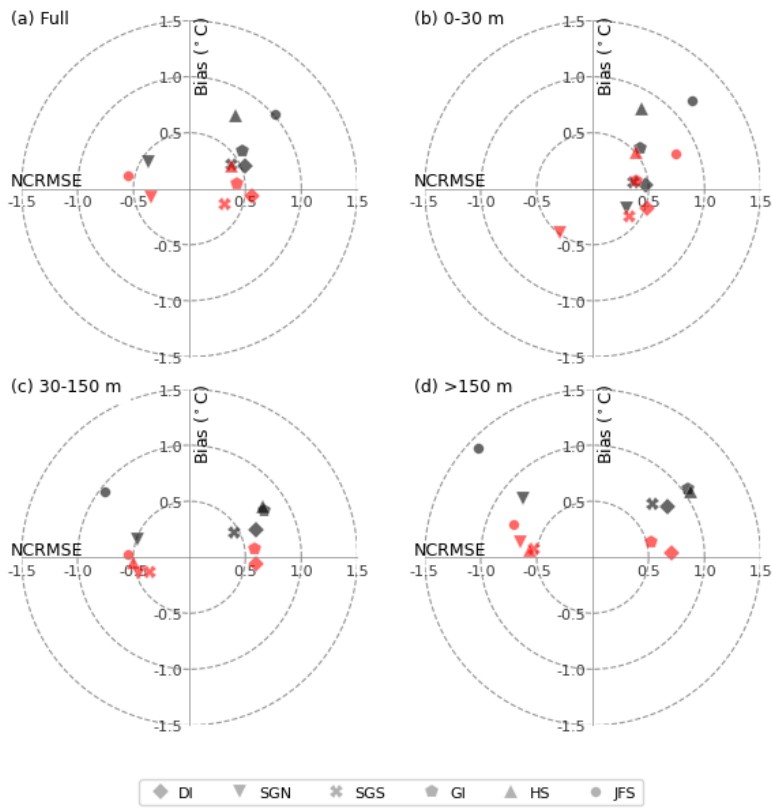

◆ DI ▼ SGN ✳ SGS ⬟ GI ▲ HS ● JFS

**Figure 6:** Target plots of the model's temperature bias and normalised centred root mean squared error (NCRMSE) using
CTD data grouped by depth strata (panels a - d) and by subdomain. Results without bias correction to western open ocean
boundary conditions (HOTSSea v1.01) are shown in grey and results with temperature bias correction (HOTSSea v1.02) are
shown in red.

The effect of the temperature bias corrections applied in HOTSSea v1.02 substantially improved
model skill versus v1.01 (Figure 5Figure 6; Tables A1 & A2). In the JFS subdomain, for example,
temperature bias calculated over all depths was reduced from +0.66 to +0.12 ℃. The HOTSSea
v1.01 run without bias correction had mean temperature biases across all depths of +0.21 to +0.66 ℃
(excluding the PS subdomain with sparse observations) versus -0.14 to +0.2 ℃ in HOTSSea v1.02.
Applying the temperature bias correction improved the model's performance in the most distant
subdomain from JFS, the DI subdomain – an unexpected result that emphasises the importance of
accurate ocean boundary forcings.





## 4.3. Model-Observation Evaluation

### 4.3.1. Vertical Profiles

Overall, the full hindcast (HOTSSea v1.02) performs best in the central and northern portion of the domain (Figure 5). The southernmost PS subdomain was omitted from analysis due to relatively few
observations collated and preparing data for this area is a priority for future work. In the SGN subdomain where the most CTD casts were available ($N$ = 12,288), the temperature bias over all depths over the hindcast period was –0.08 ℃ (0 - 30 m = -0.39 ℃, 30 - 150 m = -0.14 ℃, > 150 m = +0.13 ℃) with overall WSS of 0.97 and correlation coefficient (R) of 0.94 (Tables A1, A2). Performance was similar in the SGS subdomain. The normalised target and Taylor diagrams (Figure
6, Figures A1-A3) indicate the model captures the seasonal ocean temperature variability well, though nCRMSE was > 0.5 for JFS in the west and DI subdomains in the north. Model standard deviation is generally higher than observations in shallow depths and lower in deep water. Except for JFS and DI subdomains, the correlation coefficient with respect to temperature across all depths was above 0.9 over all depths. As depths increase, correlation generally decreases. At depths greater
than 150 m, the model underestimates temperature variability and the correlation coefficient falls to under 0.8 for DI, JFS, and SGN. All metrics are tabulated by subdomain and depth strata in Tables A1 and A2 for reference.

HOTSSea v1.02 model skill was relatively poor with respect to salinity compared with temperature,
especially in the DI and JFS subdomains (Table A2; Figures A2-A3). However, performance was better in the Strait of Georgia: mean bias over all depths in the SGS subdomain was +0.38 PSU (WSS =0.98 and R=0.96). The narrow topography in the DI combined with the 1.5 km horizontal resolution is likely leading to the observed error, as evidenced by experiments (Figure 4). Another reason for relatively poor performance with respect to salinity in areas farther from the Fraser River
could be that freshwater input from the Fraser River is forced using flow data from instruments whereas a climatology is used for other rivers in the domain (Morrison et al., 2012). The open ocean boundary conditions are forced using a climatology at the northern boundary in the DI subdomain and this would be affecting model skill. However, the latter two explanations are considered less likely given the same boundary climatology and river forcings were used here as were used in
SalishSeaCast which performs well in the DI subdomain (Figure 4).



## 4.3.2. Sea surface temperature and salinity

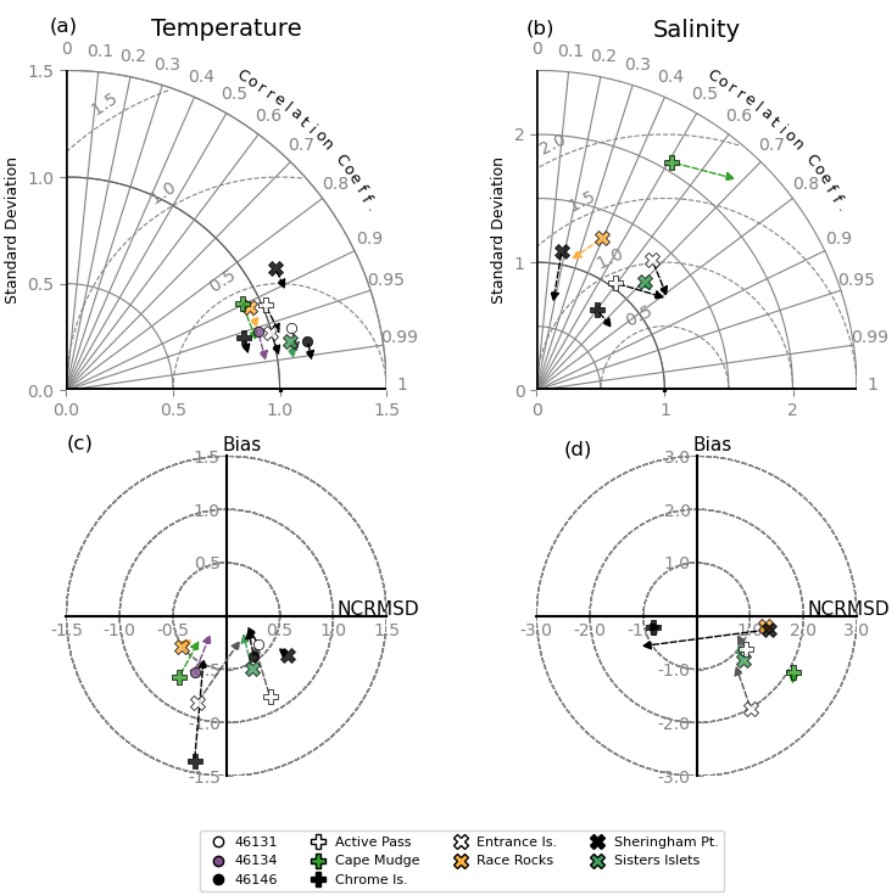

**Figure 7:** Taylor (top) and target (bottom) plots evaluating HOTSSea v1.02 sea surface temperature (SST; left) and salinity (SSS; right). SST was evaluated using data from buoys (numbered) and lightstations (named) and SSS was evaluated using lightstation data only. Statistics have been normalised (see methods) for comparison on the same plot. Perfect agreement between model and observations on the Taylor plot would correspond to normalised standard deviation = 1, correlation = 1, and NCRMSE = 0. Perfect agreement between model and observations on the target plot would be at the centre. Arrows depict the change after applying a one month moving average on both the model and observation time series.

Time series of SST and SSS taken at lightstations in the region are some of the longest in existence and present a valuable opportunity to evaluate model performance. SST and SSS were evaluated for the full hindcast using data from sampling at lightstations and buoys with temporal coverage of at least ten years. The Taylor plots indicate the model reproduces the variability in SST well (Figure 7a). Most stations have NCRMSE < 0.5 with a correlation coefficient greater than 0.9. HOTSSea v1.02 performs well at stations within the Strait of Georgia and relatively poorly Sheringham Pt in the Juan de Fuca Strait where the standard deviation of the model is approximately 20% higher than



observations. The normalised target diagram (Figure 7c), shows that the model is typically biased cold with the mean bias typically < 0.75 ℃. The one lightstation with >1 ℃ bias was Chrome Island which is located on an island close to shore where bathymetry is poorly resolved.

Overall, the model's performance is relatively high with respect to SST and low with respect to SSS.
The evaluation of SSS indicates that HOTSSea v1.02 typically overestimates variability in SSS across the domain (Figure 7b). The model standard deviation for SSS at all stations except Cape Mudge was < 1.3 times the observed standard deviation, suggesting skill at reproducing natural variability is fairly good. The correlation coefficient for SSS at most stations was relatively poor compared to SST: between 0.4 PSU and 0.7 PSU. The target plot indicates that the model is biased
fresh at the surface by approximately -2 PSU at Cape Mudge in the northern part of the domain and Entrance Island in the central Strait of Georgia. At other stations the bias was relatively low (< -1 PSU).

Many applications of HOTSSea v1 in support of ecosystem modelling and research related to Pacific salmon are anticipated to require accuracy at weekly, monthly, or seasonal time scales rather than
daily or hourly. To investigate whether the lower SSS performance was due to difficulty capturing dynamics over hourly or daily time scales versus longer time scales, we applied a monthly moving average to both the model and observations after which statistics were recalculated. Applying the moving average improved the evaluated statistics at all stations except Sheringham Point (Figure 7, arrows). The improvement leads us to conclude that if research applications require bias be < 0.5 ℃
or < 1.0 PSI then averaging results to monthly may be ensure the accuracy is acceptable with respect to SST and SSS. As shown in Sect. 4.3.1, the model performance generally improves with depth.





## 4.4. Decadal temperature trend evaluation

### 4.4.1. Nanoose station

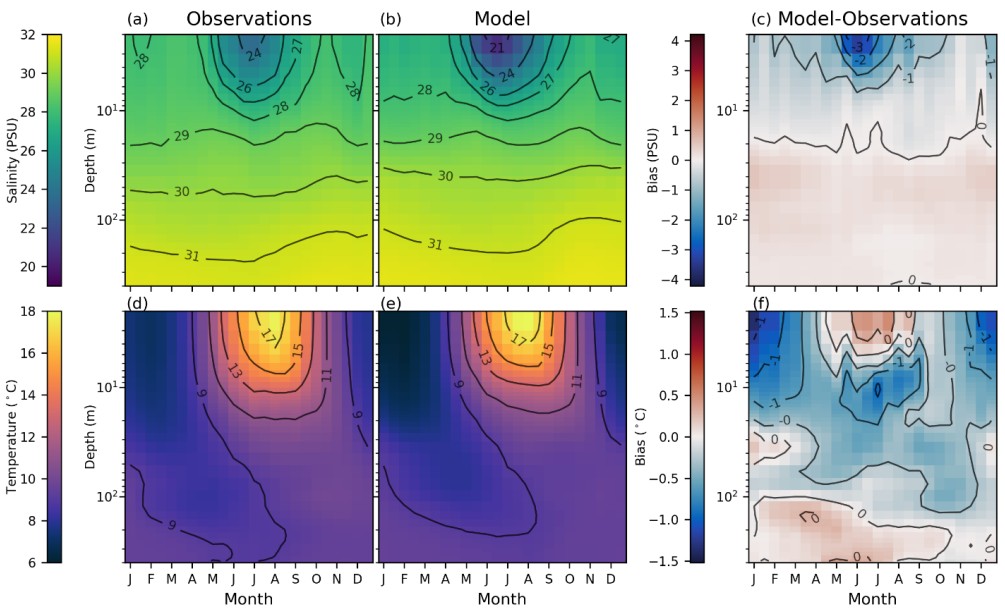

**Figure 8:** Biweekly climatologies and model-observation bias over depths over the hindcast period using measurements taken at Nanoose station between 1980 and 2018 ($N = 5,692$). The four panels to the left show salinity (top row) extracted from observations (first column) and the model (second column). Temperatures are similarly represented from observations and the model in the bottom plots. The two plots on the right show the model bias (i.e., model - observations) over depth. Data were binned bimonthly.

The model represents seasonal changes in salinity and temperature over depths well. Both the modelled and observed biweekly climatologies at Nanoose station depict a characteristic intrusion of cold water and a temperature inversion that occurs in the area in the spring and late summer or fall (Figure 8d,e), typically associated with upwelling events and neap tides (Johannessen et al., 2014; Masson, 2002; Riche et al., 2014). Strongly stratified and warm surface layers are shown in the

shallower layers in the summer in both climatologies from the observations and from the model. The model-observation biases are generally largest at depths less than 3 m for both temperature and salinity. The modelled salinity is biased fresh, especially in the summer when the bias approaches -3 PSU in depths less than 3 m. The modelled temperatures are biased warm in the summer and cool in the winter by a maximum of approximately 1 °C at depths less than 3 m. In depths greater than 10 m,

there is minor salinity bias (< 0.5 PSU). In contrast, the modelled temperatures are consistently biased slightly high (~0.5 ℃) at depths over 100 m.





After truncating the Nanoose station time series to match the model hindcast period, the monotonic temperature trend across depths 4 – 400 m was calculated using the MK test and the TS slope estimator. Significant autocorrelation was detected in the detrended biweekly time series so the 3PW

algorithm was used to adjust for this (see Sect. 3.4). The trend from observations at Nanoose station evaluated to be 0.031°C per decade (LCL: -0.018, UCL: 0.08; $p <= 0.12$). HOTSSea v1.01 *without* ocean boundary temperature bias correction detected no trend (0.00 °C / decade; LCL: -0.06, UCL: 0.06; p <= 0.73). The HOTSSea v1.02 *with* ocean temperature bias correction predicted a non-zero but insignificant trend of 0.012 °C per decade (LCL: -0.052, UCL: 0.075; $p <= 0.3$). Although trends

over the water column for the 1980 – 2018 hindcast period were not significant at the 95% confidence level, results suggested that HOTSSea v1.02 with ocean boundary temperature bias correction performs better than HOTSSea v1.01 with respect to long term-trends.

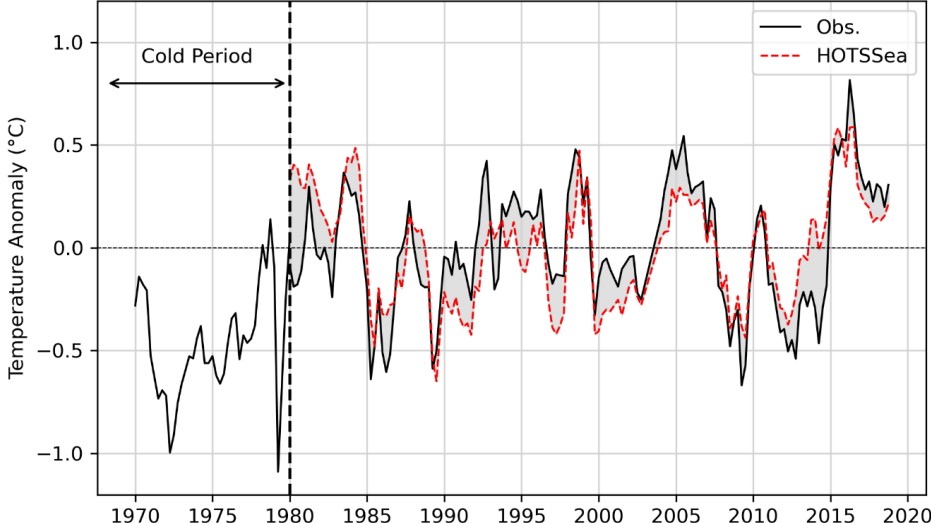

**Figure 9:** Temperature anomalies (seasonal) from observations (black, solid) at Nanoose station in the central Strait of
Georgia versus those derived from HOTSSea v1.02 model outputs (red, dashed), depth integrated over 4.5 m - 400. The grey area represents the model bias. Observations from the 1970 – 1980 period at Nanoose are included to illustrate the cooler period, a multiyear swing in temperature anomalies (1977,1978), followed by a regime shift occurring circa 1977 (Beamish et al., 1999; Hare & Mantua, 2000).

The observed inter-annual and intra-annual variability over the hindcast period is well captured by the
model (Figure **9**). The largest deviations between modelled (HOTSSea v1.02) and observed temperature anomalies over the whole water column were < 0.5 °C. Relatively large deviations from observations in the 1980 – 1983 period suggest that the one-year spin-up may be too short and this remains a priority area to investigate in the future. Warm anomalies as a result of a documented oceanic heat wave circa 2015-2016 (Gruber et al., 2021; Khangaonkar et al., 2021) are generally well

represented, suggesting the model is capable of reproducing extreme events. The evaluation of the





trends at Nanoose station also indicate the model does not incur a detectable drift in temperature bias at this location, despite being run with no data assimilation.

The secular trend previously calculated for the 1970 to 2005 period of 0.24 °C per decade (95% CI
±0.1 °C; Masson & Cummins, 2007) was over 6-fold greater than calculated here for the 1980 to 2018 hindcast period. We verified that the difference was not attributable to the methods (TS vs. LR for slope estimation) by re-estimating the trend from the observations using LR for the same 1980 – 2018 period, finding a similar result using LR (0.038 °C / decade). The weaker secular trend calculated for the 1980 – 2018 period is in part due to the omission of the 1970s, a period with cold water
temperature anomalies observed at Nanoose station (Figure 9, Figure A4). The late 1970s corresponds approximately to a polarity shift in the Pacific Decadal Oscillation (PDO; Mantua et al., 1997; Mantua & Hare, 2002), a trough in the North Pacific Gyre Oscillation (NPGO; Di Lorenzo et al., 2008), which oscillate on decadal or multi-decadal time scales. The late 1970s also corresponds to a trough in a multi-regime index circa 1977 that correlates with Northern Pacific fisheries catches
(Beamish et al., 1999). Overall, the analysis reveals a key limitation of HOTSSea v1 – omission of the 1970s – which is thus a priority area for subsequent iterations. Our analysis also serves as an update to the MC07 analysis and confirms ocean temperatures have not subsequently reverted.

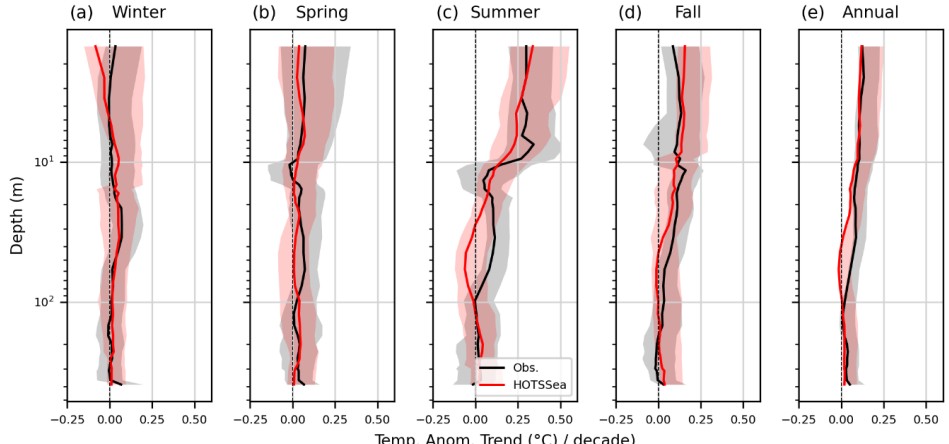

**Figure 10:** Seasonal and annual trends (1980-2018) at all depths observed at Nanoose station compared with HOTSSea v1.02 outputs. Shading represents upper and lower 95% confidence limits.

Analysis of trends at each depth at Nanoose station indicate that the modelled trends are in generally good agreement with observed trends (Figure 10). Both the modelled and observed seasonal trends in the upper water column in the summer, fall, and annually were statistically significant whereas the
trends in the winter and spring were not. A deviation between the modelled and observed trends is apparent, especially in the summer, between approximately 20 m and 100 m. The reason for the



deviation is yet to be determined but a similar deviation is also apparent in the analysis using CTDs in the GI subdomain (Figure 5). The seasonal trend patterns over depth at Nanoose station indicate that trends vary seasonally and spatially (over depths) at this location.

## 4.4.2. Strait of Georgia

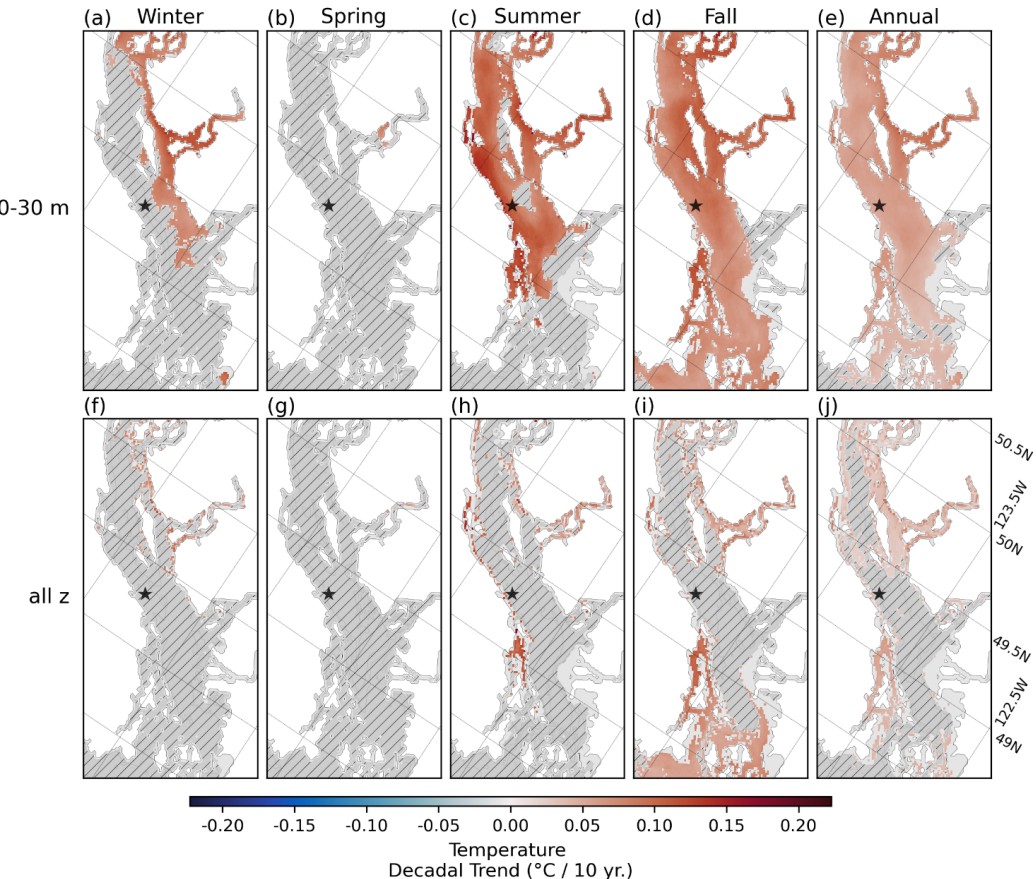

**Figure 11:** Seasonal water temperature trends in the Strait of Georgia (see Figure 2) for 0 – 30 m depth stratum and over all depths extracted from HOTSSea v1.02 model outputs. Mask (grey) has been applied to grid cells that are shallower than the depth stratum. Grey hatching has been applied to grid cells where trend was not statistically significant. Star symbol denotes
approximate location of Nanoose station.

Modelled trends from HOTSSea v1.02 were evaluated in each grid cell in the central and northern part of the model domain (Strait of Georgia and surrounding waters) over the 1980 – 2018 hindcast period (Figure 11). The 0 – 30 m depth stratum was selected given that the evaluation of the
modelled versus observed trends showed generally good agreement in approximately the top 30 m of the water column and the trends were often statistically significant (Figure 10). Although trends at





Nanoose station in the winter were not found to be statistically significant over all depths (Figure 10a), a statistically significant warming trend was detected in the 0-30 m depth stratum in the winter in the northeastern Strait of Georgia and Jervis Inlet (Figure 11a). The analysis of temperature trends in the

spring revealed no significant warming trend (Figure 11b), similar to the analysis of the Nanoose data (Figure 10b). In the summer, only the waters in the central and northern Strait of Georgia generally showed a statistically significant warming trend whereas the southern Gulf islands and surrounding waters showed no statistically significant trend (Figure 11c). Significant trends in the fall were generally widespread apart from Howe Sound (Figure 11d). Annualised trends were relatively low but

generally statistically significant throughout (Figure 11e). Note that the magnitude of all modelled trends throughout the domain are likely biased low, as the analysis of the modelled secular trend at Nanoose was weak relative to observed (see above).

## 5. Discussion and Conclusion

A contemporary challenge to achieving a long hindcast for the Salish Sea is uncertainty surrounding

the quality of the few available products to use for boundary and atmospheric forcing. The experimental approach we used involved incremental changes made to the basic setup to assess the sensitivity of model performance to different forcings using inter-model comparisons for the 2016-2018 period. The information garnered using this approach will guide future efforts to improve and extend HOTSSea v1 and possibly other models for the domain. Our analyses revealed and quantified

substantial biases inherited from both atmospheric and oceanic boundary forcings. Using the RDRS v2.1 forcings led to substantially better model performance versus using ERA5 (Sect. 4.1) - presently the only alternative with coverage of the full hindcast period. The results of our experiments suggest that locally weak winds in ERA5 resulted in reduced mixing and increased near-surface biases, leading to especially poor model performance in areas with complex and narrow topography (e.g., the

Discovery Islands). Wind speed biases in ERA5 have also been noted in other coastal and mountainous areas (Potisomporn et al., 2023). Winds are an important factor determining total productivity of the Strait of Georgia (Collins et al., 2009; Johannessen et al., 2020) and thus it was fortunate that the RDRS v2.1 atmospheric reanalysis was recently made available – using RDRS v2.1 in the final HOTSSea hindcast led to substantially better model performance. The experiments also

revealed that the global ocean reanalysis ORAS5 fields have biases at the mouth of the Juan de Fuca Strait which were inherited to some extent by the model. We suspect the main issue may be that the Salish Sea is poorly resolved in ORAS5, leading to poor representation of estuarine flow and physical dynamics at the mouth of Juan de Fuca Strait. The temperature bias correction applied here to ORAS5 at the Juan de Fuca boundary improved model performance substantially, even in areas of

the domain far from the ocean boundary. The success of our crude bias correction serves to highlight the potential benefit of relatively advanced techniques for applying bias correction such as statistical





downscaling, machine learning, or data assimilation techniques (Adachi & Tomita, 2020). Another avenue worth exploring would be to use outputs from other regional ocean hindcasts as boundary forcings, should they become available (e.g., Paquin et al., 2020; Peña et al., 2016).

Until now, gaps in long-term observations have made it difficult to confirm that the observed warming trends at Nanoose station in the Strait of Georgia are representative of trends occurring in other parts of the Salish Sea. Although an in-depth investigation into trends using HOTSSea v1 outputs was outside of the scope of this model description paper, the preliminary analysis included here highlights the model's utility for revealing the spatial-temporal aspects of decadal-scale trends. The 0 – 30 m

depth range was of particular interest because changes occurring at these depths are hypothesised to have an outstanding potential to affect dynamics of spring and fall phytoplankton blooms (Masson & Peña, 2009). HOTSSea v1 outputs indicate that the warming trend apparent at Nanoose station is not an isolated phenomenon and provide new insights into the ocean temperature trends in other parts of the domain. The model indicates the top 30 m are generally warming in the Strait of Georgia,

with some areas experiencing statistically significant warming over the entire water column (Figure 11). The fall season may also have experienced the most spatially consistent secular ocean temperature warming since 1980 over the 0 – 30 m depth range. Jervis Inlet, a fjord in the eastern portion of the domain, may be warming more rapidly than other areas, consistent with trends reported for other fjords in the region (J. M. Jackson et al., 2021). An important aspect of the model

performance is the skill at reproducing interannual variability without incurring any detectable drift, despite being a free run with no data assimilation (Figure 9Figure 10).  Anomalous events are expected to occur more frequently due to climate change and these events can lead to key biophysical thresholds being surpassed, impacting marine organisms (Gruber et al., 2021). Given that HOTSSea v1 generally replicates observed ocean temperature anomalies and extremes it thus

shows great promise for investigating pathways of effects of changes in oceanic fields due to global climate change through to marine ecosystems and to fisheries.

The performance of HOTSSea v1 assuaged initial concerns that computational cost, lower horizontal resolution, and limitations of available forcings were barriers to the development of a long hindcast for the Salish Sea. Now that the model is proven feasible, several improvements, extensions, and

applications are being explored. Secular ocean warming and increased seasonal variability have had a quantifiable effect on oxygenation of deep fjord waters in the region (J. M. Jackson et al., 2021) which may have affected the productivity or community composition of lower trophic levels (Johannessen et al., 2020). Integrating biogeochemistry into the hindcast would provide retrospective details of oxygen, nutrients, carbonate chemistry, aragonite saturation, and pH which have changed

since pre-industrial times in the Salish Sea (Jarníková et al., 2022). Coupling the physical model with a biogeochemical module would therefore be valuable. Another logical next step is to use data


assimilation to combine information from dynamical simulation with observations to produce a reanalyses, with estimates for the oceanic fields that are maximally consistent with observations (Zaron, 2011). Regarding research applications, one potentially fruitful application would be to use

velocity fields from HOTSSea v1 for Lagrangian particle simulations (Hernández-Carrasco et al., 2020; Snauffer et al., 2014). Changes to regional scale atmospheric patterns may have affected sea surface height and circulation patterns which in turn may affect important ecosystem processes such as larval dispersal. Evaluation of tides, ocean currents and the estuarine circulation would be required to support particle simulations because these are major determinants of the strength of estuarine

circulation and associated deepwater renewal in the Salish Sea (Ebbesmeyer et al., 1989; MacCready et al., 2021). The HOTSSea v1 reconstruction of temperature fields over depths should also prove useful for investigating the relative impact of pathogens and predation on fish. Warming ocean temperatures lead to greater vulnerability to pathogens and disease and thus to greater vulnerability to predation (Miller et al., 2014; Teffer et al., 2018), but concurrent increases in predator

abundances make it difficult to determine whether predation is a primary or secondary mortality factor (Walters & Christensen, 2019). Available observational time series are mainly limited to surface waters and HOTSSea v1 can help address this gap by providing ocean temperature fields at specific depths and areas occupied by fish. Regarding future iterations of the model, it remains a high priority to extend the hindcast farther backwards in time. The signal of a previously reported oceanic regime

shift occurring circa 1977 (Beamish et al., 1999; Di Lorenzo et al., 2008; Mantua et al., 1997; Mantua & Hare, 2002; MC07) is apparent in the Nanoose station time series (Figure 9). Despite various climate regime indices subsequently reverting in polarity, temperatures have not since reverted. It is thus a high priority to extend the hindcast further backwards to capture this shift once adequate atmospheric forcings are available and the previously mentioned issues with ocean boundary forcings

can be fully resolved.

Evinced by the preliminary evaluation presented here, we conclude that HOTSSea v1 shows promise for improving our understanding of long-term physical changes occurring in the Salish Sea with relevance for fish, fisheries, and marine ecosystems. The model's performance with respect to salinity, temperature, decadal scale trends, and temperature anomalies indicates HOTSSea v1 is well

suited to support ecosystem research focused on dynamics that unfold over months, seasons, years, or decades. The model addresses significant gaps in historical observations and can help drive biogeochemical and ecosystem models aimed at revealing dominant drivers of ecological productivity (Hermann et al., 2023; Macias et al., 2014; Piroddi et al., 2021).






## Author Contribution Statement

All authors contributed to the review and editing of the paper. GO wrote the original draft, conducted most data preparation, analysis, formal evaluation, and data visualisations. MD did server configuration, compilation, and running of the model with contributions from GO. TJ provided 825 expertise and scripts for trend analysis with additional revisions and development by GO. VC and MD contributed to the administration and supervision of the model development.

## Acknowledgements

Financial support for GO was provided by the Natural Sciences and Engineering Research Council of Canada (NSERC) Mitacs program (#IT12313), Fisheries and Oceans Canada, the Pacific Salmon 830 Foundation, the University of British Columbia (UBC), and the UBC Ocean Leaders Program. This research was supported by the Salish Sea Marine Survival Project. The authors thank Susan Allen and Carl Walters for review and suggestions on drafts of the manuscript and extend thanks to Elise Olson for early discussions about analysis and code. The authors also acknowledge the work of Susan Allen and colleagues who developed the SalishSeaCast model for the same domain which 835 served as the starting point for the HOTSSea v1 model.

## Code and Data Availability

HOTSSea v1 is based on the NEMO source code version 3.6 (Madec et al., 2017; subversion trunk revision rev10584), released under the open source CeCill license (https://cecill.info, last access: March 7, 2024). The standard NEMO source code can be downloaded from the NEMO website 840 (http://www.nemo-ocean.eu, last access: March 7, 2024). The data generated are 4D and are on the order of tens of terabytes when outputs are averaged to daily. The outputs are currently stored on a server without necessary bandwidth to make them publicly available; however, they can be made available by contacting the authors. The HOTSSea v1 configuration files and source code used for analysis and visuals in the present article has been archived at 845 https://doi.org/10.5281/zenodo.10846149 (Oldford, 2024).

## Competing Interests

The authors declare no competing interests.

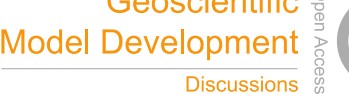

## Appendices

### A.1. - Supplemental Tables

**Table A1:** Metrics from evaluation of hindcast (HOTSSea 1.01; no temperature bias correction), grouped by subdomain and depths.

| | Depths (z) | N obs. | Temperature: HOTSSea v1.01 | | | | | Temperature: HOTSSea v1.02 | | | | |
|---|---|---|---|---|---|---|---|---|---|---|---|---|
| | | | Model | Bias | RMSE | R | WSS | Model | Bias | RMSE | R | WSS |
| DI | all z | 3649 | 10.06 | 0.2 | 0.68 | 0.89 | 0.93 | 9.74 | -0.06 | 0.63 | 0.88 | 0.93 |
| | 0->30 m | 3640 | 10.48 | 0.03 | 0.84 | 0.9 | 0.95 | 10.24 | -0.17 | 0.85 | 0.9 | 0.94 |
| | 30->150 m | 3083 | 9.72 | 0.25 | 0.51 | 0.87 | 0.9 | 9.41 | -0.06 | 0.45 | 0.88 | 0.92 |
| | >150 m | 1118 | 9.74 | 0.45 | 0.55 | 0.79 | 0.75 | 9.34 | 0.04 | 0.33 | 0.76 | 0.86 |
| SGN | all z | 12365 | 9.73 | 0.24 | 0.47 | 0.93 | 0.95 | 9.4 | -0.08 | 0.37 | 0.94 | 0.97 |
| | 0->30 m | 12349 | 10.36 | -0.17 | 0.63 | 0.95 | 0.98 | 10.13 | -0.39 | 0.7 | 0.96 | 0.97 |
| | 30->150 m | 11409 | 9.26 | 0.16 | 0.33 | 0.89 | 0.92 | 8.95 | -0.14 | 0.32 | 0.9 | 0.92 |
| | >150 m | 8348 | 9.64 | 0.52 | 0.59 | 0.8 | 0.66 | 9.25 | 0.13 | 0.31 | 0.77 | 0.83 |
| SGS | all z | 4140 | 9.79 | 0.21 | 0.56 | 0.95 | 0.97 | 9.42 | -0.14 | 0.42 | 0.95 | 0.97 |
| | 0->30 m | 4138 | 10.36 | 0.06 | 0.69 | 0.95 | 0.97 | 10.08 | -0.25 | 0.65 | 0.95 | 0.97 |
| | 30->150 m | 3437 | 9.47 | 0.22 | 0.4 | 0.94 | 0.95 | 9.13 | -0.13 | 0.32 | 0.94 | 0.96 |
| | >150 m | 1136 | 9.61 | 0.48 | 0.57 | 0.86 | 0.79 | 9.21 | 0.08 | 0.31 | 0.85 | 0.91 |
| GI | all z | 2512 | 10.61 | 0.34 | 0.87 | 0.93 | 0.95 | 10.27 | 0.04 | 0.7 | 0.94 | 0.96 |
| | 0->30 m | 2512 | 11.09 | 0.36 | 0.89 | 0.94 | 0.95 | 10.77 | 0.07 | 0.73 | 0.94 | 0.97 |
| | 30->150 m | 1832 | 10.01 | 0.41 | 0.8 | 0.92 | 0.9 | 9.68 | 0.08 | 0.6 | 0.93 | 0.94 |
| | >150 m | 175 | 9.26 | 0.61 | 0.84 | 0.7 | 0.73 | 8.78 | 0.13 | 0.38 | 0.86 | 0.92 |
| HS | all z | 800 | 10.45 | 0.65 | 0.88 | 0.96 | 0.93 | 10.01 | 0.2 | 0.59 | 0.97 | 0.97 |
| | 0->30 m | 800 | 10.96 | 0.71 | 0.95 | 0.96 | 0.92 | 10.58 | 0.33 | 0.66 | 0.97 | 0.96 |
| | 30->150 m | 610 | 9.63 | 0.45 | 0.66 | 0.84 | 0.84 | 9.13 | -0.05 | 0.37 | 0.87 | 0.93 |
| | >150 m | 442 | 9.22 | 0.59 | 0.79 | 0.67 | 0.68 | 8.69 | 0.06 | 0.35 | 0.83 | 0.9 |
| JFS | all z | 3707 | 9.29 | 0.66 | 0.92 | 0.74 | 0.75 | 8.74 | 0.12 | 0.47 | 0.85 | 0.91 |
| | 0->30 m | 3707 | 10.38 | 0.79 | 1.18 | 0.79 | 0.77 | 9.9 | 0.31 | 0.79 | 0.81 | 0.87 |
| | 30->150 m | 3521 | 8.91 | 0.59 | 0.84 | 0.69 | 0.72 | 8.35 | 0.03 | 0.43 | 0.84 | 0.91 |
| | >150 m | 448 | 8.06 | 0.98 | 1.16 | 0.36 | 0.47 | 7.37 | 0.29 | 0.52 | 0.72 | 0.79 |
| PS | all z | 99 | 11.12 | -0.1 | 0.8 | 0.89 | 0.94 | 10.81 | -0.41 | 0.91 | 0.89 | 0.92 |
| | 0->30 m | 99 | 11.62 | 0.04 | 0.73 | 0.92 | 0.96 | 11.32 | -0.25 | 0.77 | 0.92 | 0.95 |
| | 30->150 m | 88 | 10.69 | -0.25 | 0.94 | 0.85 | 0.91 | 10.36 | -0.57 | 1.09 | 0.84 | 0.88 |
| | >150 m | 3 | 11.66 | -0.15 | 0.43 | 0.6 | 0.71 | 11.18 | -0.63 | 0.75 | 0.57 | 0.55 |



**Table A2:** Metrics from evaluation of hindcast (HOTSSea v1.02; temperature bias correction) over the hindcast period (1980 – 2018), grouped by subdomain and depths.

| | Depths (z) | N obs. | Salinity: HOTSSea v1.01 | | | | | Salinity: HOTSSea v1.02 | | | | |
|---|---|---|---|---|---|---|---|---|---|---|---|---|
| | | | Model Mean | Bias | RMSE | R | WSS | Model Mean | Bias | RMSE | R | WSS |
| DI | all z | 3474 | 29.11 | -0.19 | 1.06 | 0.72 | 0.79 | 29.28 | -0.09 | 1.03 | 0.68 | 0.75 |
| | 0->10 m | 3465 | 27.75 | -0.56 | 1.18 | 0.46 | 0.64 | 27.87 | -0.48 | 1.14 | 0.4 | 0.61 |
| | 30->150 m | 3062 | 29.91 | 0.2 | 0.84 | 0.65 | 0.67 | 29.96 | 0.25 | 0.85 | 0.66 | 0.67 |
| | >150 m | 1118 | 30.82 | 0.22 | 0.43 | 0.81 | 0.7 | 30.87 | 0.26 | 0.46 | 0.81 | 0.68 |
| SGN | all z | 12288 | 30.25 | 0.29 | 0.49 | 0.92 | 0.94 | 30.33 | 0.34 | 0.49 | 0.93 | 0.93 |
| | 0->10 m | 12272 | 28.3 | 0.03 | 0.69 | 0.71 | 0.83 | 28.38 | 0.09 | 0.67 | 0.72 | 0.83 |
| | 30->150 m | 11388 | 30.5 | 0.41 | 0.48 | 0.77 | 0.66 | 30.56 | 0.47 | 0.52 | 0.79 | 0.62 |
| | >150 m | 8347 | 31.2 | 0.26 | 0.3 | 0.63 | 0.57 | 31.25 | 0.3 | 0.34 | 0.64 | 0.52 |
| SGS | all z | 3834 | 28.57 | 0 | 2.73 | 0.89 | 0.93 | 29.23 | 0.38 | 1.45 | 0.96 | 0.98 |
| | 0->10 m | 3832 | 27.29 | 0.05 | 2.8 | 0.86 | 0.91 | 27.84 | 0.44 | 1.6 | 0.95 | 0.97 |
| | 30->150 m | 3437 | 30.56 | 0.46 | 0.5 | 0.86 | 0.69 | 30.61 | 0.51 | 0.55 | 0.86 | 0.65 |
| | >150 m | 1136 | 31.21 | 0.32 | 0.35 | 0.85 | 0.71 | 31.26 | 0.37 | 0.39 | 0.86 | 0.66 |
| GI | all z | 2473 | 29.02 | -0.58 | 1.21 | 0.67 | 0.77 | 29.1 | -0.53 | 1.18 | 0.66 | 0.76 |
| | 0->10 m | 2473 | 28.44 | -0.65 | 1.27 | 0.56 | 0.67 | 28.51 | -0.6 | 1.25 | 0.56 | 0.67 |
| | 30->150 m | 1832 | 29.79 | -0.46 | 0.78 | 0.76 | 0.79 | 29.84 | -0.41 | 0.75 | 0.77 | 0.8 |
| | >150 m | 175 | 31.54 | -0.14 | 0.49 | 0.91 | 0.93 | 31.57 | -0.1 | 0.48 | 0.91 | 0.93 |
| HS | all z | 800 | 30.49 | -0.2 | 0.69 | 0.94 | 0.93 | 30.54 | -0.16 | 0.67 | 0.94 | 0.93 |
| | 0->10 m | 800 | 29.6 | -0.47 | 0.81 | 0.87 | 0.85 | 29.64 | -0.43 | 0.78 | 0.87 | 0.86 |
| | 30->150 m | 610 | 31.32 | 0.15 | 0.41 | 0.71 | 0.82 | 31.37 | 0.2 | 0.43 | 0.71 | 0.81 |
| | >150 m | 442 | 31.99 | 0.1 | 0.35 | 0.83 | 0.9 | 32.03 | 0.14 | 0.36 | 0.84 | 0.9 |
| JFS | all z | 3675 | 32.18 | -0.02 | 0.51 | 0.82 | 0.9 | 32.23 | 0.02 | 0.48 | 0.83 | 0.91 |
| | 0->10 m | 3675 | 30.73 | -0.52 | 0.93 | 0.42 | 0.59 | 30.79 | -0.46 | 0.88 | 0.41 | 0.6 |
| | 30->150 m | 3514 | 32.7 | 0.19 | 0.43 | 0.87 | 0.92 | 32.73 | 0.22 | 0.45 | 0.87 | 0.91 |
| | >150 m | 448 | 33.83 | 0.07 | 0.21 | 0.78 | 0.86 | 33.84 | 0.07 | 0.21 | 0.78 | 0.86 |
| PS | all z | 99 | 29.7 | -0.15 | 0.9 | 0.55 | 0.69 | 29.77 | -0.08 | 0.88 | 0.56 | 0.7 |
| | 0->10 m | 99 | 29.11 | -0.48 | 1.04 | 0.43 | 0.59 | 29.18 | -0.4 | 0.99 | 0.45 | 0.61 |
| | 30->150 m | 88 | 30.13 | 0.05 | 0.73 | 0.61 | 0.73 | 30.2 | 0.12 | 0.72 | 0.62 | 0.74 |
| | >150 m | 3 | 30.88 | 0.19 | 0.24 | 1 | 0.86 | 30.96 | 0.28 | 0.3 | 1 | 0.79 |

A.2. – Supplemental Figures

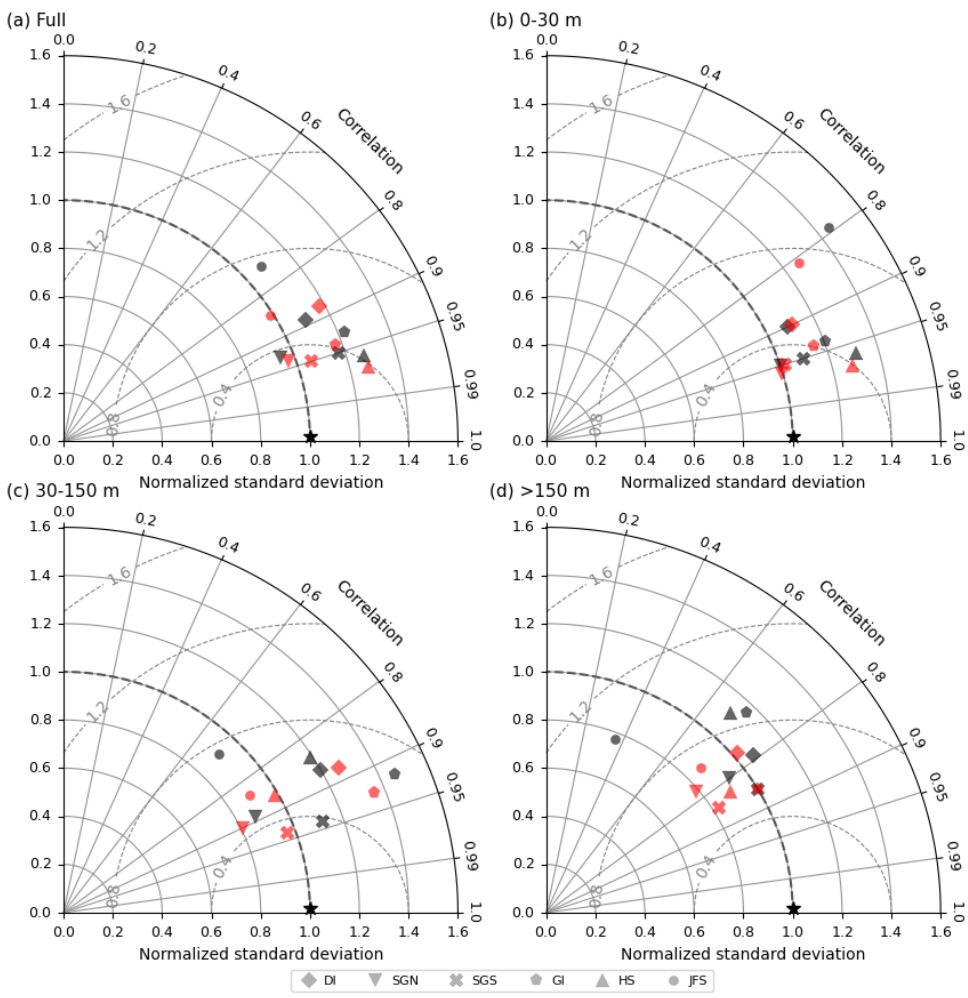

**Figure A1:** Taylor plots of temperature using CTD measurements grouped by subdomain and illustrating change in model performance after applying temperature bias correction at open boundary (grey = HOTSSea v1.01; before bias correction, red = HOTSSea v1.02; after bias correction). Standard deviation (solid grey contours) and centred root mean square error (dashed grey contours) have been normalised.




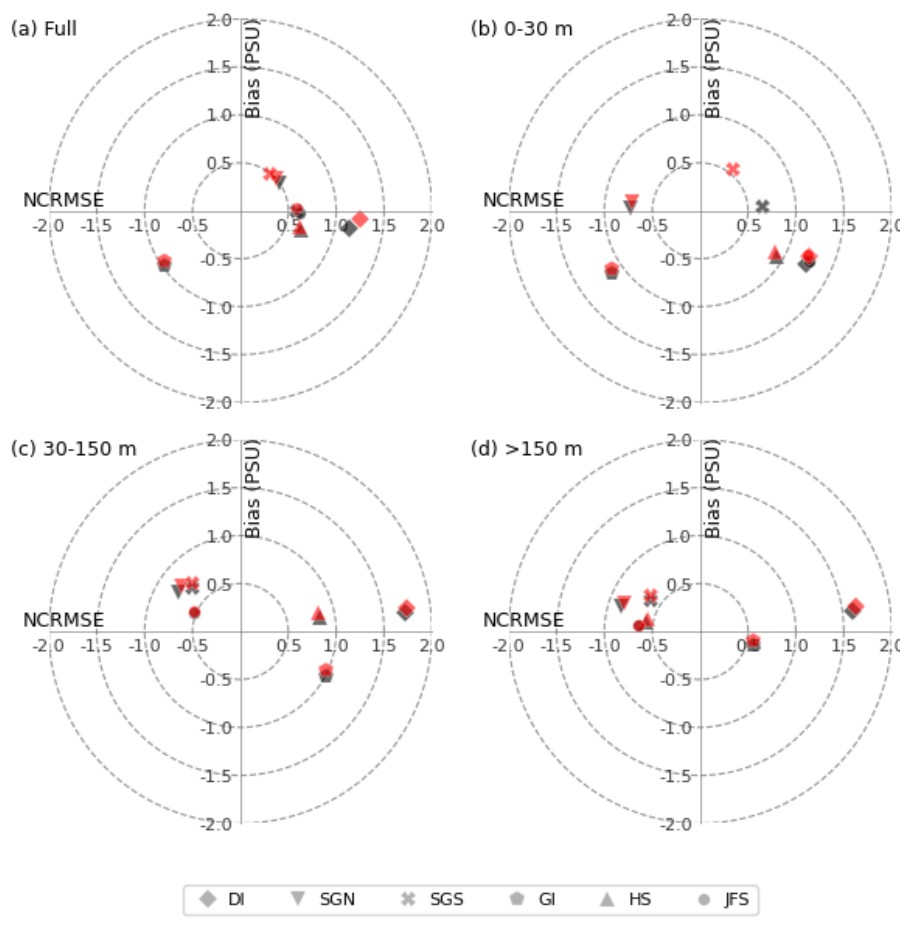


**Figure A2:** Target plots of the model's salinity bias and normalised centred root mean squared error (NCRMSE) using CTD data grouped by depth strata (panels a - d) and by subdomain. Results without bias correction to western open ocean boundary conditions (HOTSSea v1.01) are shown in grey and results with temperature bias correction (HOTSSea v1.02) are shown in red.




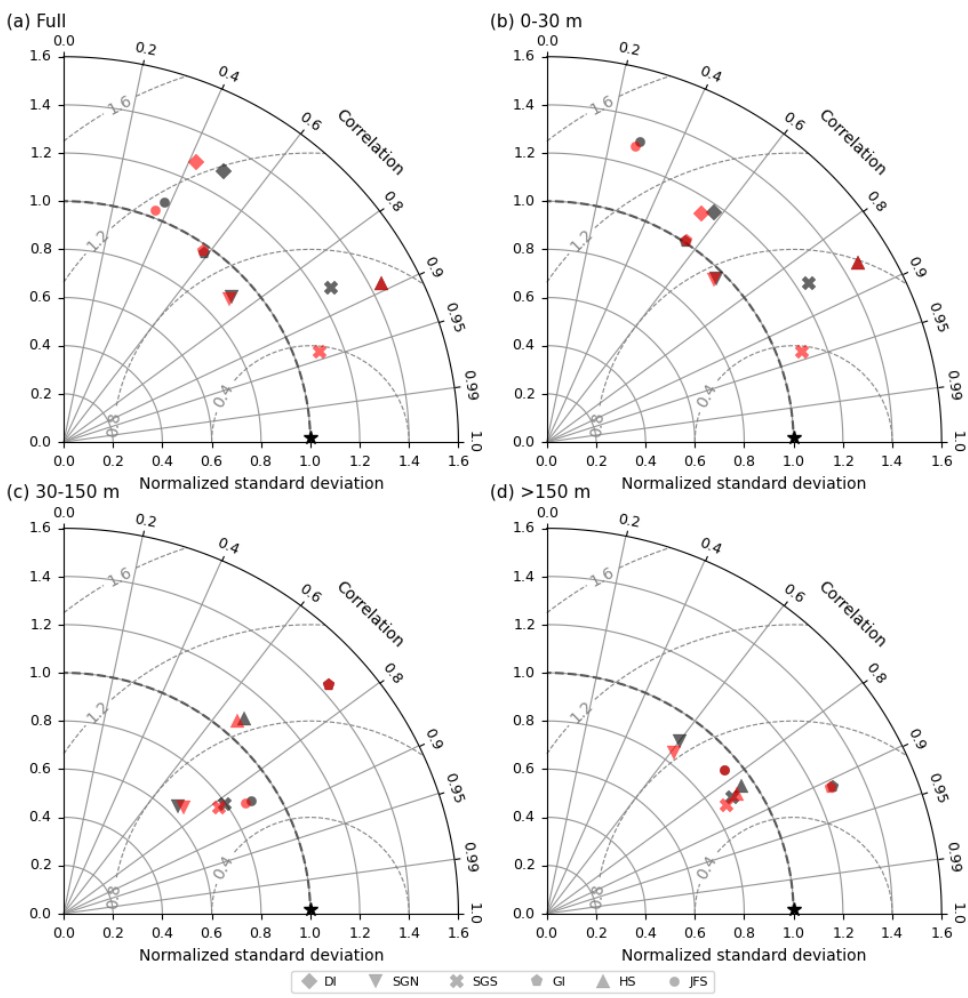

**Figure A3:** Taylor diagrams showing salinity (PSU) using CTD measurement grouped by depth and subdomain. Plots show minor differences in temperature bias correction at open boundary before bias correction (grey = HOTSSea v1.01) versus after bias correction (red = HOTSSea v1.02). Standard deviation (solid grey contours) and centred root mean square error (dashed grey contours) have been normalised.






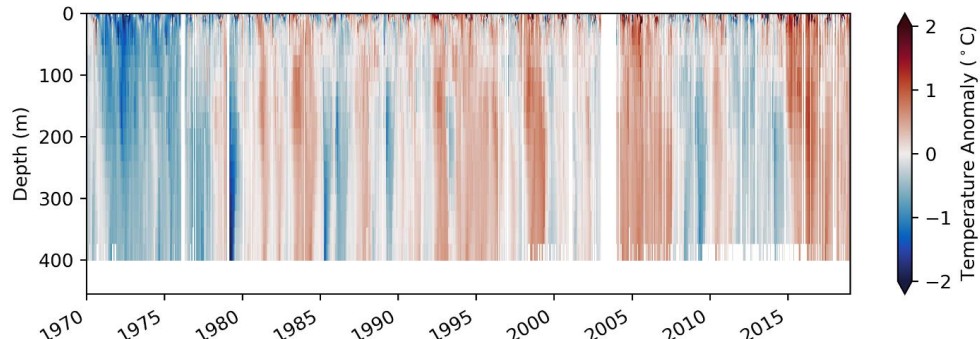

**Figure A4:** Water temperature anomalies from observations taken at Nanoose station from 1970 to present illustrating the cold 1970 – 1979 period not covered by the model hindcast.






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
