# Peer review of "HOTSSea v1: a NEMO-based physical Hindcast of the Salish Sea (1980 – 2018) supporting ecosystem model development"

_Geoscientific Model Development, 2024_

## Referee Comment (RC1)

**GMD-2024-58**

**Review of "HOTSSea v1: a NEMO-based physical Hindcast of the Salish Sea (1980 – 2018) supporting ecosystem model development" (GMD-2024-58)**

**1. GMD Scope and Aims:**

- **Scope and Aim**:

    - The paper falls within the scope of GMD, as it focuses on the development and evaluation of a numerical model (HOTSSea v1) for simulating the physical oceanography of the Salish Sea.

    - The aim of the paper is to present a long-term (1980-2018) hindcast model that can address the lack of observational data and support ecosystem model development in the region.

    - The paper is relevant to the GMD readership, as it addresses challenges and provides insights related to regional ocean modeling, forcing data selection and bias correction, and the application of models to understanding long-term changes and variability in marine systems

- **Relevance:**

    - The Salish Sea is a complex and ecologically important region facing significant pressures from climate change and human activities. The development of a reliable hindcast model is crucial for understanding the region's physical dynamics and their impacts on marine ecosystems and fisheries.
    - The paper's focus on the selection and evaluation of forcing data is highly relevant, as the accuracy of forcing data is a major factor influencing the performance of regional ocean models.
    - The bias correction method, while simple, is effective in improving model performance and highlights the importance of addressing biases in forcing data.
    - The model's ability to reproduce observed temperature variability and trends, as well as extreme events like marine heatwaves, demonstrates its potential for studying climate change impacts and informing ecosystem-based management.
    - The paper contributes to the broader field of ocean modeling by providing insights into the challenges and strategies for developing and evaluating long-term hindcast models in complex coastal systems.
    - While the paper addresses important research questions, its relevance could be further strengthened by explicitly discussing how the model could be used to answer specific scientific questions related to climate change impacts, ecosystem dynamics, and fisheries management in the Salish Sea.
    - The authors could also elaborate on the model's potential limitations and uncertainties, particularly regarding its performance in underrepresented regions (e.g., Puget Sound) and the impact of biases on specific ecological processes.

- **Scientific Significance:**
    - Strengths:
        - Addresses a critical need for a long-term hindcast model in the Salish Sea, a region with limited observational data and facing significant ecological challenges.
        - Provides a valuable baseline for assessing future climate change impacts and developing ecosystem-based management strategies.
        - Offers insights into the spatial and temporal patterns of ocean warming, which are crucial for understanding potential impacts on marine ecosystems and fisheries.
        - Demonstrates the importance of bias correction in regional ocean modeling and the potential effectiveness of even simple correction methods.
    - Weaknesses:
        - The study's scope could be broadened by explicitly addressing specific scientific questions related to climate change, ecosystem dynamics, and fisheries management that the model could help answer.
        - The limitations of the model, particularly in terms of spatial coverage and salinity bias, could impact its applicability for addressing certain research questions.
- **Originality:**
    - Strengths
        - Addresses a clear gap in existing research by providing the first long-term (38-year) hindcast model of the Salish Sea, which is crucial for understanding long-term changes and variability.
        - Employs a systematic experimental approach to assess the sensitivity of the model to different forcing data sets, offering insights into the impact of forcing choices on model performance.
        - Presents new findings on spatial and temporal patterns of ocean warming in the Salish Sea, particularly the potential for faster warming in Jervis Inlet compared to other areas.
        - Demonstrates the feasibility of using a relatively coarse resolution model with bias correction to produce a useful hindcast for ecosystem studies
    - Weaknesses:
        - While the model itself is novel for the Salish Sea, the underlying NEMO model and bias correction methods are established techniques. The paper could further emphasize the unique aspects of the model setup or application that distinguish it from previous work.
        - The authors could discuss the potential implications of their findings for other regional ocean modeling studies, particularly those in complex coastal environments with limited observational data.

**2. Scientific Quality:**

- **Soundness:**
    - Strengths:

- Model Setup: The authors provide a detailed description of the model configuration, including spatial and temporal resolution, numerical schemes, and parameterizations. This allows for reproducibility and assessment of the model's suitability for the Salish Sea.
- Forcing Data: The paper thoroughly discusses the selection and evaluation of forcing data, highlighting the challenges of using reanalysis products and the importance of bias correction. This transparency is commendable and aids in understanding the potential limitations of the model.
- Experimental Design: The authors employ a systematic approach to assess the model's sensitivity to different forcing data sets, providing valuable insights into the sources of error and bias.
- Bias Correction: The implementation of a temperature bias correction, while simple, demonstrates a clear effort to improve the model's accuracy. The authors acknowledge the limitations of their method and suggest potential future improvements.
- Weaknesses:
  - NEMO Version: The use of NEMO v3.6 is outdated, as newer versions offer potential improvements in numerical schemes and physical parameterizations. The authors could be asked to discuss the potential benefits of upgrading to a more recent version.
  - Simplified Tidal Forcing: The use of only eight tidal constituents and outdated WebTide data might not fully capture the complex tidal dynamics of the Salish Sea, potentially impacting the accuracy of the model's circulation and mixing.
  - River Discharge Data: The reliance on climatological river discharge data, except for the Fraser River, could introduce uncertainties in the model's representation of freshwater inputs and their effects on salinity.
  - Single Station for Trend Analysis: The use of only one station (Nanoose) for evaluating long-term trends might not be sufficient to capture the spatial variability of changes in the Salish Sea. The authors could be encouraged to explore the use of additional observational data or alternative methods for trend analysis.
- **Completeness**:

  - Strengths:
    - Model Description: The paper provides a thorough description of the model setup, including spatial and temporal resolution, numerical schemes, parameterizations, and forcing data. This allows for reproducibility and assessment of the model's suitability for the Salish Sea.
    - Experimental Design: The authors clearly outline their experimental approach, including the different model versions and the rationale for varying the forcing data. This systematic approach helps to isolate the sources of error and bias in the model.
    - Results Presentation: The paper presents a comprehensive set of results, including statistical metrics, figures, and tables, providing a detailed picture of the model's performance in simulating temperature, salinity, trends, and anomalies.

- Discussion of Limitations: The authors openly acknowledge the limitations of their study, such as the exclusion of Puget Sound, and the potential for further improvements in bias correction. This transparency is commendable and adds to the completeness of the paper.
  - Weaknesses:
    - Lack of Sensitivity Analysis: While the authors discuss the sensitivity of the model to different forcing data sets, a more quantitative sensitivity analysis could further strengthen their conclusions. This could involve systematically varying model parameters or forcing inputs to assess their impact on the results.
    - Limited Discussion of Physical Mechanisms: The paper could benefit from a more in-depth discussion of the physical mechanisms responsible for the observed trends and biases. This would provide deeper insights into the model's behavior and help to identify areas for further improvement.
    - Comparison with Other Models: A more comprehensive comparison with other existing models for the Salish Sea would help to contextualize the results and highlight the unique contributions of HOTSSea v1. This could involve comparing model outputs, skill scores, or the ability to reproduce specific observed features.
    - Data and Code Availability: While the authors mention that the data and code are available upon request, making them publicly accessible would enhance transparency and reproducibility, facilitating further research and model development by the wider community.

**3. Presentation Quality:**

- Clarity:

  - Strengths:
    - Overall Structure: The paper is well-structured, with clear sections and sub-sections that guide the reader through the model development, evaluation, and discussion.
    - Logical Flow: The narrative follows a logical progression, starting with the introduction and motivation, then describing the model setup and forcing data, followed by a detailed evaluation of the model's performance, and concluding with a discussion of potential future directions.
    - Clear Figures and Tables: Most figures and tables are well-organized and effectively convey the key results of the study. Figure 1 provides a clear overview of the model domain, and the Taylor diagrams (Figures 4, 7) offer a concise summary of model performance.
  - Weaknesses:
    - Terminology: Some acronyms (e.g., CRMSE, WSS) are not explicitly defined upon first use, which could confuse readers unfamiliar with the specific terminology.
    - Equation Presentation: Some equations could be presented more clearly, with better explanations of the variables and symbols used. For example, Equation 1 for the Coriolis parameter could be clarified with additional context and definition of terms.

- Figure Captions: Some figure captions could be more informative and self-contained, providing sufficient context to understand the figure without having to refer back to the main text.
- Redundancy: The text could be more concise in some sections, as there is some repetition of information and excessive detail in certain parts of the methods and results.

- **Conciseness:**

  - Strengths:

    - Key Points Highlighted: The paper generally focuses on the essential aspects of the model development, evaluation, and discussion, highlighting the key findings and implications for Salish Sea research.
    - Relevant Literature Review: The introduction provides a concise overview of the relevant literature, focusing on the importance of the Salish Sea and the need for improved oceanographic models.
    - Effective Use of Tables and Figures: The authors make good use of tables (e.g., Table 1 summarizing forcing data) and figures (e.g., Taylor diagrams) to present information in a concise and visually appealing manner.

  - Weaknesses:

    - Repetitive Information: There is some repetition of information across different sections, particularly in the results and discussion sections. For example, the poor performance of the model in the Puget Sound subdomain is mentioned multiple times. w
    - Excessive Detail: Some sections, especially the methods section, contain excessive detail that could be condensed or moved to supplementary material. For instance, the detailed description of the statistical tests could be streamlined or summarized.
    - Lengthy Sentences: Some sentences are overly long and complex, making it difficult for the reader to follow the authors' train of thought. Breaking these sentences into shorter, more focused ones would improve readability.

  - Recommendations:

    - Eliminate Redundancy: Carefully review the text and eliminate any unnecessary repetition of information. Consider consolidating similar findings or moving less critical details to supplementary material.
    - Streamline Methods: Condense the description of standard methods (e.g., statistical tests) and focus on the specific choices and adaptations made for this study.

- Illustrations:

  - Strengths:
    - Informative Figures: The figures generally provide valuable information and support the main text effectively. For example, Figure 1 clearly depicts the model domain and bathymetry, while Figures 4 and 7 summarize the model's performance in a concise and visually appealing manner.
    - Adequate Number of Figures: The number of figures seems appropriate for the length and complexity of the paper. They are distributed

throughout the text to illustrate key concepts and results, aiding in the reader's understanding.

- Appropriate Figure Types: The authors use a variety of figure types (maps, time series plots, Taylor diagrams, target plots) that are well-suited for presenting different types of data and results.
- Color Schemes and Clarity: The figures are clear and easy to interpret, with appropriate use of color schemes that effectively highlight the key features of the data.
  - o Weaknesses:
    - Caption Detail: While most figure captions adequately describe the content of the figures, some could be more informative and self-contained. For instance, the captions for Figure 4 , more details on the interpretation of diagrams, However Figure 7 does explain the target plot and taylor plot.

**4. Open Science Considerations:**

- Model and Data Availability:

  - o Strengths:
    - The authors state that the model code and output data are available upon request, demonstrating a willingness to share their research materials.
    - They have made their custom analysis package available online, which promotes transparency and facilitates reproducibility for certain aspects of the analysis.
- Weaknesses:

  - o Code Availability: The model code itself is not publicly available, which is a significant limitation for a paper published in GMD. This hinders reproducibility and prevents other researchers from fully scrutinizing and building upon the authors' work.
  - o Data Accessibility: While the data are available upon request, this process can be cumbersome and may limit access for some researchers. Making the data publicly available in a repository with a persistent identifier (e.g., DOI) would be more in line with open science principles.
- Note with regards to Editor's Messages on the above:

  - o The editor's messages highlight the importance of adhering to GMD's data and code policy. They emphasize that making the code available is a requirement for publication and that simply stating "available upon request" is not sufficient. The editor also questions the appropriateness of using GitHub repositories for code storage and suggests that referencing them in the paper might be problematic.
- Recommendations (Considering Editor's Messages):

- Make the Model Code Publicly Available: The authors should make their model code publicly available in a suitable repository (e.g., Zenodo, institutional repository) with a persistent identifier. This is essential for reproducibility and transparency.

- Clarify Data Availability: While the authors state that the data are available upon request, they should clarify the process for obtaining the data and consider making it publicly

available in a repository with a DOI. This would facilitate access for other researchers and promote open science practices.

- Additional Considerations:

- Licensing: The authors should ensure that the model code and data are released under an open-source license that allows for reuse and modification by others. This would further enhance the impact and reach of their research.

- Documentation: The authors could provide additional documentation (e.g., README files, user manuals) to explain the model's setup, input requirements, and output formats. This would make it easier for others to use and adapt the model for their own research purposes.

**5. Overall Assessment and Recommendation:**

- Summary of Strengths:

  - Addresses a Critical Gap: The paper successfully addresses a significant gap in long-term observational data for the Salish Sea, providing a much-needed resource for understanding decadal-scale changes in physical oceanography.
  - Model Skill: The HOTSSea v1.02 model demonstrates good skill in reproducing observed temperature variability and trends, especially after the implementation of the temperature bias correction at the Juan de Fuca Strait boundary.
  - Bias Correction Effectiveness: The simple bias correction method employed by the authors is shown to be effective in reducing the warm bias in the ORAS5 forcing data, leading to improved model performance.
  - Experimental Design: The systematic approach used to assess the sensitivity of the model to different forcing datasets is a strength, as it provides valuable insights into the sources of error and bias.
  - Relevance to Ecosystem Modeling: The paper clearly articulates the relevance of the model for ecosystem modeling, highlighting its potential to drive biogeochemical and ecosystem models aimed at understanding ecological productivity in the Salish Sea.
  - Potential for Future Applications: The authors outline several promising future applications of the model, including biogeochemical modeling, data assimilation, and Lagrangian particle simulations, demonstrating the model's broader utility beyond the scope of this study
- Summary of Weaknesses:

  - Limited Spatial Coverage:
    - The model's evaluation is primarily focused on the Strait of Georgia, with limited assessment in other regions, particularly the Puget Sound. This raises concerns about the generalizability of the results to the entire Salish Sea.
  - Persistent Biases:
    - The model exhibits persistent biases in salinity, especially in the Discovery Islands and Juan de Fuca Strait. The fresh bias in these regions could significantly impact the accuracy of ecosystem models that rely on this data.

- While the temperature bias correction is effective, there are still depth-dependent biases in temperature, particularly in deeper waters, indicating potential limitations in the model's representation of vertical processes.
  - Limited Validation Data:
    - The reliance on a single station (Nanoose) for trend analysis and evaluation of interannual variability limits the robustness of the conclusions. The authors should consider using additional data sources (e.g., satellite data, other stations) to validate the model's performance across a wider range of conditions.
  - Simplified Tidal Forcing:
    - The use of only eight tidal constituents and outdated WebTide data might not fully capture the complex tidal dynamics of the Salish Sea, potentially impacting the accuracy of the model's circulation and mixing.
  - Outdated NEMO Version:
    - The use of NEMO v3.6, an older version of the model, might limit the model's capabilities and potential for improvement. Newer versions could offer enhanced numerical schemes and physical parameterizations.
  - Uncertainty Quantification:
    - The paper lacks a rigorous quantification of uncertainties associated with the model's predictions. This makes it difficult to assess the reliability of the results and their implications for future research and applications.

5. Overall Assessment and Recommendation

- Recommendation:

  - I recommend this paper for publication in GMD after minor revisions. The paper presents a valuable contribution to the field of regional ocean modeling by developing a much-needed hindcast model for the Salish Sea. The model demonstrates good skill in reproducing observed temperature variability and trends, and the authors' systematic approach to evaluating forcing data and bias correction is commendable. However, the limitations related to spatial coverage, salinity bias, and depth-dependent errors warrant further attention and should be more thoroughly addressed before final publication.
- Justification:

  - The authors have successfully developed a 38-year hindcast model of the Salish Sea, which is a significant achievement given the limited observational data available for this region. The model's ability to reproduce observed temperature patterns, particularly after bias correction, suggests that it can be a valuable tool for understanding long-term changes and variability in the Salish Sea. The authors' transparency in discussing the limitations of the model, such as its relatively coarse resolution and the lack of data assimilation, is commendable.

  - However, the model's performance in simulating salinity remains a concern, especially in the Discovery Islands and Juan de Fuca Strait. The depth-dependent biases in temperature also raise questions about the model's ability to accurately represent vertical processes. Additionally, the reliance on a single station for trend analysis and the limited spatial coverage of the evaluation warrant further investigation.

- To address these limitations, the authors should consider:

    - Expanding the spatial coverage of the evaluation to include Puget Sound and other underrepresented regions.
    - Investigating the sources of the salinity bias and exploring alternative bias correction methods or model refinements to address this issue.
    - Analyzing the model's performance at finer temporal resolutions relevant to ecological processes.
    - Quantifying uncertainties associated with the model's predictions and conducting sensitivity analyses to assess the robustness of the results.
    - Exploring the potential benefits of data assimilation to further improve model accuracy.
    - The authors could also consider moderating their conclusions regarding the model's applicability for ecosystem modeling, given the lack of data assimilation and the remaining biases. While the model shows promise for studying long-term changes, its limitations should be acknowledged and addressed before it can be confidently used for ecological applications.

Overall, this paper is a valuable contribution to the field of regional ocean modeling, but it requires minor revisions to strengthen its conclusions and address its limitations.

**6. Detailed Comments:**

- Abstract:

    - Strengths:
        - Concisely summarizes the key points of the paper, including the motivation, methodology, main findings, and potential applications.
        - Clearly states the problem of observational gaps in the Salish Sea and the need for a long-term hindcast model.
        - Highlights the model's skill in reproducing observed temperature trends and anomalies, as well as its potential for supporting ecosystem model development.
    - Weaknesses:
        - Could be more specific about the magnitude and types of biases found in the forcing data and the model.
        - The statement about "new insights" into ocean trends could be more explicit, outlining what specific new information the model provides.
        - Could briefly mention the limitations of the model, such as the exclusion of Puget Sound and the persistent bias in salinity.
- Section 1

    - Strengths:
        - Provides a comprehensive overview of the Salish Sea's ecological and economic importance, emphasizing the need for better understanding of its physical oceanography.
        - Clearly articulates the motivation for developing a long-term hindcast model, citing the lack of observational data and the potential impacts of climate change on the ecosystem.

- Discusses the limitations of existing models for the Salish Sea, justifying the need for a new model like HOTSSea v1.
- Outlines the specific objectives of the paper, focusing on evaluating the model's performance and investigating potential biases.
  - Weaknesses:
  - The discussion of climate change impacts on the Salish Sea could be more specific, mentioning specific examples of observed or projected changes in temperature, salinity, or circulation patterns.
  - The potential applications of the model for ecosystem modeling could be more clearly articulated, with specific examples of how the model's output could be used to inform ecological research and management.
- Section 2. Model Overview:

  - NEMO Version: While the use of NEMO 3.6 is understandable given resource constraints, the authors should acknowledge that newer versions (e.g., NEMO 4.x ) offer potential advantages, such as improved numerical schemes, physical parameterizations, wetting and drying and computational efficiency. A brief discussion of these potential benefits and the rationale for using NEMO 3.6 would strengthen the paper.

  - Model Resolution: The choice of 1.5 km horizontal resolution is a compromise between accuracy and computational cost. However, the authors should discuss the implications of this resolution for capturing fine-scale processes, particularly in areas with complex topography like the Discovery Islands. A sensitivity analysis with different resolutions could be considered in future work.

  - Vertical Coordinates: The use of z-level coordinates might not be ideal for the Salish Sea, which is characterized by steep bathymetry and strong stratification. The authors could discuss the potential advantages of using terrain-following (s- or sigma-) coordinates, which might better capture vertical variations in water properties.

  - Section 2.2 Boundary Conditions and Forcings:

    - Open Boundary Conditions: The use of a monthly climatology for the northern boundary and outdated WebTide data for tidal forcing could introduce errors in the model. The authors should acknowledge these limitations and discuss potential alternatives, such as using higher-resolution reanalysis data or a more recent tidal model.

    - River Discharge: The reliance on climatological river discharge data for most rivers (except the Fraser River) could also introduce uncertainties, as it doesn't account for interannual variability. The authors could explore using a hydrological model to provide more realistic and temporally varying freshwater inputs.

- Section 3. Model Evaluation:

  - Limited Spatial Coverage: The evaluation primarily focuses on the Strait of Georgia, with limited assessment in other regions, especially Puget Sound. This raises concerns about the generalizability of the results. The authors should acknowledge this limitation and discuss potential strategies for expanding the evaluation in future work.

- o Choice of Metrics: While the use of CRMSE and WSS is appropriate, the authors could consider additional metrics to provide a more comprehensive evaluation of model performance. For example, they could assess the model's skill in reproducing specific features of the Salish Sea circulation, such as the estuarine exchange flow or the seasonal cycle of stratification.
  - o Statistical Significance: The authors should consistently report the statistical significance of their results, particularly for trend analysis and comparisons between different model versions. This would strengthen the robustness of their conclusions.
- Section 4. Results and Discussion:

  - o Depth-Dependent Biases: The model exhibits biases in both shallow and deep waters, particularly for temperature. The authors should delve deeper into the potential causes of these biases, such as inaccuracies in the forcing data, limitations in the vertical mixing scheme, or other model parameterizations.
  - o Salinity Bias: The persistent fresh bias in salinity, especially in the Discovery Islands and Juan de Fuca Strait, remains a major concern. The authors should investigate the sources of this bias and explore potential model refinements or alternative forcing data to address it.
  - o Overestimation of Variability: The model tends to overestimate the variability in salinity. This issue warrants further investigation to understand its potential causes and implications for ecosystem modeling.
- Section 5. Conclusions:

- Overconfidence: The authors express a high degree of confidence in the model's capabilities despite the identified limitations. They should acknowledge these limitations more explicitly and moderate their conclusions accordingly.

- Future Work: The authors should provide a more detailed roadmap for future model development, including plans to address the identified limitations, explore alternative forcing data and bias correction methods, and expand the model's evaluation to include other regions and variables.

---

## Referee Comment (RC3)

**General feedback:**

**SalishSeaCast**

The use of SalishSeaCast in your evaluations as opposed to solely relying on limited observations is a great choice. However, I feel that relying so heavily on a model without discussing its limitations and how they could relate to your interpretation of HOTSSea's effectiveness could make your evaluations somewhat misleading. Please include a discussion of SalishSeaCast limitations before presenting comparisons to it.

**Python packages**

I wouldn't normally reference which python packages are used (ex. lines 113-119), I think that giving access to your code and referencing it at some point (as you have done) is sufficient. It is also difficult to imagine how a package is used without looking at the notebook/script, so I don't think that this information adds much to your paper. Adding a README to your zenodo and GitHub repository, pointing to which folders one should look into for specific parts of the paper (such as temperature bias correction or SalishSeaCast evaluations), may be more useful to those interested in applying your methods.

**Statistical equations**

The equation for widely known statistical metrics do not need to be detailed in the text (equations 1 and 2 for example), just mention which were used to keep the paper concise. It is much more important to describe what the metric reveals about a dataset. Where lesser-known metrics were used, or modifications to traditional ones were made, then it makes sense to keep the equations.

**Puget Sound Evaluation**

It is necessary to draw the line somewhere, and you explain in the outline of future work that the collation of observations in Puget Sound is a necessary next step. However, since a large portion of your evaluations rely on SalishSeaCast, which covers Puget Sound, it is not clear why Puget Sound was not included in that analysis. I think that the paper would benefit from either the inclusion of this work or a justification for why it was left out. Stressing that hindcasts of Puget Sound have already been done, as you mention on line 81, may do the trick.

**Line-by-line comments:**

**Line 56 –** Feely et al., (2010) and Ianson et al., (2016) are also worthwhile regional acidification papers to reference here.

**Line 253 –** I found this part confusing, explicitly state the difference between v0.14 and v0.16 to increase clarity.

**Table 3 –** I went back to this table many times while reading the text, here are a few suggestions to increase clarity:

- Are HRDPS1 and HRDPS different? If not, keep naming consistent.
- I think you are missing the "H" in "RDPS2 (10 km)"
- I liked the addition of approximate resolution next to HRDPS1 and RDPS in the second row. Include this for all the forcings.
- In your "evaluation purpose" for v0.16 I think that adding that this run went back to the "full" atmospheric forcing resolution would increase clarity on the difference between v0.14 and v0.16.

**Figure 2 –** Add Nanoose station (the back star) to the legend.

**Table 4 –** Normalize the CTD count in each subdomain by area in order to discuss heterogeneity more accurately in **line 274**.

**Line 287, 293, 297 –** Define acronyms.

**Lines 302-309 –** This belongs earlier, after line 282 perhaps, where you describe the model depth and time indexing. Tell us (the reader) about what you're conducting statistics on before telling us about the statistical metrics.

**Line 315 –** Accidental paragraph break?

**Line 324 –** I can't find this dataset in Table 2. If I am not mistaken, then why not include it? If you do include it, then I don't believe that the sentence "Canadian buoy data were downloaded…" is needed.

**Line 351 –** "Described above" at the beginning of a new section is confusing, specify which section it was described in.

**Line 367 –** Remove "it"

**Line 382 –** I also don't see Nanoose station in table 2, I'm a bit confused about why some observations were not included.

**Line 393 –** Does "depth strata of the closest HOTSSea grid cell" refer to the depth strata described on line 304 or the depth range of the closest grid cell?

 **Lines 434-436 –** Very cool!

**Line 447 –** I'm not familiar with this statistic – how large is "large"?

**Line 494 –** Worthwhile to remind the reader of the difference between v0.12 and v0.14 to reveal what this similarity tells you.

**Line 497 –** Confusing wording since v0.14 and v0.16 both use ORAS5.

**Lines 499-505 –** Make this description more clearly comparative between v0.14 and v0.16, at times it is unclear which version you're referring to.

**Line 508 –** Perhaps I'm mistaken (and if I am than consider rewording) but I read this line as the higher resolution forcing (HRDPS) leading to lower bias than the lower resolution forcing (ERA5). I don't understand why this was unexpected, please expand.

**Line 510 –** I interpret "across all depths" to mean here that the bias was lower everywhere. However, later in the paper it seems to mean the average over the whole water column. Make sure not to use this phrase for both applications.

**Figure 5 –** Impressive changes after bias correction!

**Line 576 –** nCRMSE is defined with a capital "N" earlier in the paper.

**Figure 7a-7b –** Add NCRMSE label to the plots.

**Figure 7c-7d –** NCRMSE instead of NCRMSD

**Lines 615-618 –** These sentences seem to contradict. Is SSS variability overestimated or is it good?

**Line 623 –** In addition to a discussion of the time resolution required for HOTSSea to support ecosystem modelling this paper needs a discussion of what spatial resolution is required. Explain to the reader why 1.5 km is good enough.

**Line 630 –** PSU?

**Line 630 –** remove "be"

**Line 648 –** This seems like a rather large salinity bias at the surface to me. Put it into context. What is it in SalishSeaCast? Do you think that this surface salinity bias will affect model circulation?

**Line 689 –** How does it correlate?

**Figure 11 –** Could there be a better way to express in the caption that it is a seasonal average of decadal trends? I think solely calling it "seasonal" in the caption could be a bit confusing.

**Line 750 –** Is it your intension to conduct a similar bias correction on salinity despite the instability problems you mentioned on line 369?

**Line 771 –** Missing space between Figure 9 and Figure 10

**Table A1 –** Something cutoff under "Model" in the top row

**Figure A3 –** Just a note that I think that the description of these Taylor diagrams is more clear than what you have in the main body

---

## Author Comment (AC2)

Hello,

As requested, we have revised our digital assets to comply with the GMD data and code policy. We created a new Github code repo and therefore have new Zenodo DOI's and links to provide, which are shown below.
I don't see a way to revise the assets and links using the online portal. Please let me know the best way to do so.

################
Data sets:
Forcings 1: HRDPS
https://zenodo.org/doi/10.5281/zenodo.12193923
Forcings 2: RDRS
https://zenodo.org/doi/10.5281/zenodo.12206290
Forcings 3: ERA5, CIOPS, Runoff, etc
https://zenodo.org/doi/10.5281/zenodo.12312768
Note - due to storage limits, three Zenodo archives were required

################
Model code and software:
HOTSSea_v1_NEMOandSupportCode
https://doi.org/10.5281/zenodo.12520931
which archives the following github repo:
https://github.com/goldford/HOTSSea_v1_NEMOandSupportCode/tree/main
with files specifically requested by the reviewer and editor here (NEMO sources):
https://github.com/goldford/HOTSSea_v1_NEMOandSupportCode/tree/main/serverside/NEMO

################
Interactive computing environment:
HOTSSea_v1_NEMOandSupportCode
https://doi.org/10.5281/zenodo.12520931
which archives the following github repo:
https://github.com/goldford/HOTSSea_v1_NEMOandSupportCode/tree/main
with python notebooks found here:
https://github.com/goldford/HOTSSea_v1_NEMOandSupportCode/tree/main/desktop/code
Note - this is in the same repo as the model code and software

Best wishes,
Greig Oldford

-----Original Message-----
From: editorial@copernicus.org <editorial@copernicus.org>
Sent: Friday, June 14, 2024 3:57 AM

To: Oldford, Greig <Greig.Oldford@dfo-mpo.gc.ca>
Cc: editor@mailarchive.copernicus.org
Subject: gmd-2024-58 - Chief editor comment posted

Dear Greig Oldford,

We are pleased to inform you that Juan Antonio Añel posted a new Chief editor comment in the interactive discussion of the following GMD preprint:

Title: HOTSSea v1: a NEMO-based physical Hindcast of the Salish Sea (1980–2018) supporting ecosystem model development
Author(s): Greig Oldford et al.
MS No.: gmd-2024-58
MS type: Model description paper

Please access the discussion at:
https://can01.safelinks.protection.outlook.com/?url=https%3A%2F%2Fgmd.copernicus.org%2Fpreprints%2Fgmd-2024-58%2F%23discussion&data=05%7C02%7Cgreig.oldford%40dfo-mpo.gc.ca%7C2eead9324a6b460955a508dc8c60baf6%7C1594fdaea1d94405915d011467234338%7C0%7C0%7C638539594292527157%7CUnknown%7CTWFpbGZsb3d8eyJWIjoiMC4wLjAwMDAiLCJQIjoiV2luMzIiLCJBTiI6Ik1haWwiLCJXVCI6Mn0%3D%7C0%7C%7C%7C&sdata=WMllJGaCYzm4DkxnfmUhQIsn%2B6fJ%2FCiAdjo4upb8nT0%3D&reserved=0

To log in, please use your Copernicus Office user ID 752190.

In case any questions arise, please do not hesitate to contact me.

Kind regards,

The editorial support team
Copernicus Publications
editorial@copernicus.org

Data protection remark: emails sent from the online system CO Editor might include the email address editor@mailarchive.copernicus.org as CC. When replying to such emails and keeping this email address in CC, your replies will be archived in the online system CO Editor alongside the manuscript identified through the manuscript number in the subject line. Such archived emails are accessible for the respective handling editors, journal's chief editors, as well as Copernicus Publications' staff members.

---

## Author Comment (AC7)

The authors wish to extend their gratitude to the anonymous reviewer. The manuscript was substantially improved based on their helpful feedback. Please see our responses in red below.

**Referee 1: Review of "HOTSSea v1: a NEMO-based physical Hindcast of the Salish Sea (1980 – 2018) supporting ecosystem model development" (GMD-2024-58)**

**1. GMD Scope and Aims:**

- **Scope and Aim**:

  - The paper falls within the scope of GMD, as it focuses on the development and evaluation of a numerical model (HOTSSea v1) for simulating the physical oceanography of the Salish Sea.

  - The aim of the paper is to present a long-term (1980-2018) hindcast model that can address the lack of observational data and support ecosystem model development in the region.

  - The paper is relevant to the GMD readership, as it addresses challenges and provides insights related to regional ocean modeling, forcing data selection and bias correction, and the application of models to understanding long-term changes and variability in marine systems

- **Relevance:**

  - The Salish Sea is a complex and ecologically important region facing significant pressures from climate change and human activities. The development of a reliable hindcast model is crucial for understanding the region's physical dynamics and their impacts on marine ecosystems and fisheries.
  - The paper's focus on the selection and evaluation of forcing data is highly relevant, as the accuracy of forcing data is a major factor influencing the performance of regional ocean models.
  - The bias correction method, while simple, is effective in improving model performance and highlights the importance of addressing biases in forcing data.
  - The model's ability to reproduce observed temperature variability and trends, as well as extreme events like marine heatwaves, demonstrates its potential for studying climate change impacts and informing ecosystem-based management.
  - The paper contributes to the broader field of ocean modeling by providing insights into the challenges and strategies for developing and evaluating longterm hindcast models in complex coastal systems.
  - While the paper addresses important research questions, its relevance could be further strengthened by explicitly discussing how the model could be used to answer specific scientific questions related to climate change impacts, ecosystem dynamics, and fisheries management in the Salish Sea. Thank you for this suggestion, we have now added discussion about how the model could specifically help, with examples. (e.g., intro, third paragraph)

- o The authors could also elaborate on the model's potential limitations and uncertainties, particularly regarding its performance in underrepresented regions (e.g., Puget Sound) and the impact of biases on specific ecological processes. We have now added discussion about the limitation of the model with respect to Puget sound and how biases may affect specific ecological processes. (e.g., Line 829)
- **Scientific Significance:**
  - o Strengths:
    - Addresses a critical need for a long-term hindcast model in the Salish Sea, a region with limited observational data and facing significant ecological challenges.
    - Provides a valuable baseline for assessing future climate change impacts and developing ecosystem-based management strategies.
    - Offers insights into the spatial and temporal patterns of ocean warming, which are crucial for understanding potential impacts on marine ecosystems and fisheries.
    - Demonstrates the importance of bias correction in regional ocean modeling and the potential effectiveness of even simple correction methods.
  - Weaknesses:
    - The study's scope could be broadened by explicitly addressing specific scientific questions related to climate change, ecosystem dynamics, and fisheries management that the model could help answer. Though 'the scope could be broadened' is true for any study, we have now added discussion with examples of specific scientific questions. Overall, the present study's scope was deliberately limited to yield a manuscript of appropriate length for GMD.
    - The limitations of the model, particularly in terms of spatial coverage and salinity bias, could impact its applicability for addressing certain research questions. (see comments now, throughout – e.g., line 597)
- **Originality:**
  - o Strengths
    - Addresses a clear gap in existing research by providing the first long-term (38-year) hindcast model of the Salish Sea, which is crucial for understanding long-term changes and variability.
    - Employs a systematic experimental approach to assess the sensitivity of the model to different forcing data sets, offering insights into the impact of forcing choices on model performance.
    - Presents new findings on spatial and temporal patterns of ocean warming in the Salish Sea, particularly the potential for faster warming in Jervis Inlet compared to other areas.
    - Demonstrates the feasibility of using a relatively coarse resolution model with bias correction to produce a useful hindcast for ecosystem studies
  - ⑩ Weaknesses:
    - While the model itself is novel for the Salish Sea, the underlying NEMO model and bias correction methods are established techniques. The paper

could further emphasize the unique aspects of the model setup or application that distinguish it from previous work.

- We now emphasise some of the unique aspects of the work in the last paragraph of the intro and in the discussion (e.g., line 768). To our knowledge there are no other studies that looked at atmospheric forcing resolution on a Salish Sea model, nor have they run a Salish Sea hindcast with the recently-available forcing products (RDRS 2.1, ORAS5).
- The authors could discuss the potential implications of their findings for other regional ocean modeling studies, particularly those in complex coastal environments with limited observational data. We believe this is now addressed throughout (e.g., implicit from Lines 809 – 811).

**2. Scientific Quality:**

- **Soundness:**

  o Strengths:

  Model Setup: The authors provide a detailed description of the model configuration, including spatial and temporal resolution, numerical schemes, and parameterizations. This allows for reproducibility and assessment of the model's suitability for the Salish Sea.

  - Forcing Data: The paper thoroughly discusses the selection and evaluation of forcing data, highlighting the challenges of using reanalysis products and the importance of bias correction. This transparency is commendable and aids in understanding the potential limitations of the model.
  - Experimental Design: The authors employ a systematic approach to assess the model's sensitivity to different forcing data sets, providing valuable insights into the sources of error and bias.
  - Bias Correction: The implementation of a temperature bias correction, while simple, demonstrates a clear effort to improve the model's accuracy. The authors acknowledge the limitations of their method and suggest potential future improvements.

  o Weaknesses:

  - NEMO Version: The use of NEMO v3.6 is outdated, as newer versions offer potential improvements in numerical schemes and physical parameterizations. The authors could be asked to discuss the potential benefits of upgrading to a more recent version. We now have noted that it is a priority in the future to upgrade to NEMO v4.0 (Line 847)
  - Simplified Tidal Forcing: The use of only eight tidal constituents and outdated WebTide data might not fully capture the complex tidal dynamics of the Salish Sea, potentially impacting the accuracy of the model's circulation and mixing.
    The 8 constituents capture the bulk of the tidal energy on the west coast. CIOPS-W and SSC use the same eight. We state that WebTide was the starting point and further tuning was applied to the tides. NEMO is responsible for capturing the tidal dynamics, not WebTide. The reviewer describes WebTide as outdated but we think there is no reason to believe the tides have changed since WebTide was produced.

- River Discharge Data: The reliance on climatological river discharge data, except for the Fraser River, could introduce uncertainties in the model's representation of freshwater inputs and their effects on salinity.
- Single Station for Trend Analysis: The use of only one station (Nanoose) for evaluating long-term trends might not be sufficient to capture the spatial variability of changes in the Salish Sea. The authors could be encouraged to explore the use of additional observational data or alternative methods for trend analysis. We make notes throughout that Nanoose is indeed not sufficient – yet it is the only station with data; filling the gaps elsewhere and examining heterogeneity is a key motivator for this work. This should now be clearer in our revised intro and discussion.

- **Completeness**:

  - Strengths:
    - Model Description: The paper provides a thorough description of the model setup, including spatial and temporal resolution, numerical schemes, parameterizations, and forcing data. This allows for reproducibility and assessment of the model's suitability for the Salish Sea.

    - Experimental Design: The authors clearly outline their experimental approach, including the different model versions and the rationale for varying the forcing data. This systematic approach helps to isolate the sources of error and bias in the model.

    - Results Presentation: The paper presents a comprehensive set of results, including statistical metrics, figures, and tables, providing a detailed picture of the model's performance in simulating temperature, salinity, trends, and anomalies.

    - Discussion of Limitations: The authors openly acknowledge the limitations of their study, such as the exclusion of Puget Sound, and the potential for further improvements in bias correction. This transparency is commendable and adds to the completeness of the paper.

  Weaknesses:
    - Lack of Sensitivity Analysis: While the authors discuss the sensitivity of the model to different forcing data sets, a more quantitative sensitivity analysis could further strengthen their conclusions. This could involve systematically varying model parameters or forcing inputs to assess their impact on the results. While this would be a nice addition, it would expand the scope beyond what is reasonable for a single, focused article. We also argue that our experimental evaluation does serve as a basic sensitivity analysis (with respect to forcings).
    - Limited Discussion of Physical Mechanisms: The paper could benefit from a more in-depth discussion of the physical mechanisms responsible for the observed trends and biases. This would provide deeper insights into

the model's behavior and help to identify areas for further improvement. Our discussion was limited to the effect of physical forcings, as mechanisms leading to error and bias. We feel that expanding the discussion to focus on various mechanisms would be nice, but outside the scope of this model description paper.

- Comparison with Other Models: A more comprehensive comparison with other existing models for the Salish Sea would help to contextualize the results and highlight the unique contributions of HOTSSea v1. This could involve comparing model outputs, skill scores, or the ability to reproduce specific observed features. We feel it was reasonable to select SSC as the model to compare our outputs to, given our research questions (see last paragraph of intro).
- Data and Code Availability: While the authors mention that the data and code are available upon request, making them publicly accessible would enhance transparency and reproducibility, facilitating further research and model development by the wider community. This has been addressed.

**3. Presentation Quality:**

- Clarity:
  - Strengths:
    - Overall Structure: The paper is well-structured, with clear sections and sub-sections that guide the reader through the model development, evaluation, and discussion.
    - Logical Flow: The narrative follows a logical progression, starting with the introduction and motivation, then describing the model setup and forcing data, followed by a detailed evaluation of the model's performance, and concluding with a discussion of potential future directions.
    - Clear Figures and Tables: Most figures and tables are well-organized and effectively convey the key results of the study. Figure 1 provides a clear overview of the model domain, and the Taylor diagrams (Figures 4, 7) offer a concise summary of model performance.
  - Weaknesses:
    - Terminology: Some acronyms (e.g., CRMSE, WSS) are not explicitly defined upon first use, which could confuse readers unfamiliar with the specific terminology. This has now been addressed.
    - Equation Presentation: Some equations could be presented more clearly, with better explanations of the variables and symbols used. For example, Equation 1 for the Coriolis parameter could be clarified with additional context and definition of terms. We have made edits to ensure all symbols and variables are defined. We are not sure what Coriolis parameter you are referring to.
    - Figure Captions: Some figure captions could be more informative and self-contained, providing sufficient context to understand the figure without having to refer back to the main text. Edits to several captions have been made.

- ▪ Redundancy: The text could be more concise in some sections, as there is some repetition of information and excessive detail in certain parts of the methods and results. Thank you, we have tried to be more concise by making edits throughout.

- **Conciseness:**
  - o Strengths:
    - ▪ Key Points Highlighted: The paper generally focuses on the essential aspects of the model development, evaluation, and discussion, highlighting the key findings and implications for Salish Sea research.
    - ▪ Relevant Literature Review: The introduction provides a concise overview of the relevant literature, focusing on the importance of the Salish Sea and the need for improved oceanographic models.
    - ▪ Effective Use of Tables and Figures: The authors make good use of tables (e.g., Table 1 summarizing forcing data) and figures (e.g., Taylor diagrams) to present information in a concise and visually appealing manner.

    Weaknesses:
    - ▪ Repetitive Information: There is some repetition of information across different sections, particularly in the results and discussion sections. For example, the poor performance of the model in the Puget Sound subdomain is mentioned multiple times. To be clear, we state throughout that Puget Sound remains essentially unevaluated, with rationale given.
    - ▪ Excessive Detail: Some sections, especially the methods section, contain excessive detail that could be condensed or moved to supplementary material. For instance, the detailed description of the statistical tests could be streamlined or summarized. Some details (e.g., equations) have been reduced.
    - ▪ Lengthy Sentences: Some sentences are overly long and complex, making it difficult for the reader to follow the authors' train of thought. Breaking these sentences into shorter, more focused ones would improve readability. We have made changes throughout to help improve readability in this respect.

  - o Recommendations:
    - ▪ Eliminate Redundancy: Carefully review the text and eliminate any unnecessary repetition of information. Consider consolidating similar findings or moving less critical details to supplementary material. This has been done.
    - ▪ Streamline Methods: Condense the description of standard methods (e.g., statistical tests) and focus on the specific choices and adaptations made for this study. The standard methods have been condensed and we have removed unnecessary equations for well-known statistics.

- Illustrations:
  - o Strengths:
    - ▪ Informative Figures: The figures generally provide valuable information and support the main text effectively. For example, Figure 1 clearly depicts

the model domain and bathymetry, while Figures 4 and 7 summarize the model's performance in a concise and visually appealing manner.

- Adequate Number of Figures: The number of figures seems appropriate for the length and complexity of the paper. They are distributed throughout the text to illustrate key concepts and results, aiding in the reader's understanding.
- Appropriate Figure Types: The authors use a variety of figure types (maps, time series plots, Taylor diagrams, target plots) that are well-suited for presenting different types of data and results.
- Color Schemes and Clarity: The figures are clear and easy to interpret, with appropriate use of color schemes that effectively highlight the key features of the data.

- Weaknesses:
  - Caption Detail: While most figure captions adequately describe the content of the figures, some could be more informative and self contained. For instance, the captions for Figure 4 , more details on the interpretation of diagrams, However Figure 7 does explain the target plot and Taylor plot. Thank you, the captions have been revised.

**4. Open Science Considerations:**

- Model and Data Availability:

  - Strengths:
    - The authors state that the model code and output data are available upon request, demonstrating a willingness to share their research materials.
    - They have made their custom analysis package available online, which promotes transparency and facilitates reproducibility for certain aspects of the analysis.
- Weaknesses:

  - Code Availability: The model code itself is not publicly available, which is a significant limitation for a paper published in GMD. This hinders reproducibility and prevents other researchers from fully scrutinizing and building upon the authors' work.
  - Data Accessibility: While the data are available upon request, this process can be cumbersome and may limit access for some researchers. Making the data publicly available in a repository with a persistent identifier (e.g., DOI) would be more in line with open science principles.
- Note with regards to Editor's Messages on the above:

  - The editor's messages highlight the importance of adhering to GMD's data and code policy. They emphasize that making the code available is a requirement for publication and that simply stating "available upon request" is not sufficient. The editor also questions the appropriateness of using GitHub repositories for code storage and suggests that referencing them in the paper might be problematic. This has now been addressed.
- Recommendations (Considering Editor's Messages):

- Make the Model Code Publicly Available: The authors should make their model code publicly available in a suitable repository (e.g., Zenodo, institutional repository) with a persistent identifier. This is essential for reproducibility and transparency. This has now been addressed.

- Clarify Data Availability: While the authors state that the data are available upon request, they should clarify the process for obtaining the data and consider making it publicly available in a repository with a DOI. This would facilitate access for other researchers and promote open science practices. This has been addressed.

- Additional Considerations:

- Licensing: The authors should ensure that the model code and data are released under an open-source license that allows for reuse and modification by others. This would further enhance the impact and reach of their research. This has been addressed.

- Documentation: The authors could provide additional documentation (e.g., README files, user manuals) to explain the model's setup, input requirements, and output formats. This would make it easier for others to use and adapt the model for their own research purposes. This has been addressed, a README is included.

**5. Overall Assessment and Recommendation:**

- Summary of Strengths:

  - Addresses a Critical Gap: The paper successfully addresses a significant gap in long-term observational data for the Salish Sea, providing a much-needed resource for understanding decadal-scale changes in physical oceanography. ○ Model Skill: The HOTSSea v1.02 model demonstrates good skill in reproducing observed temperature variability and trends, especially after the implementation of the temperature bias correction at the Juan de Fuca Strait boundary.
  - Bias Correction Effectiveness: The simple bias correction method employed by the authors is shown to be effective in reducing the warm bias in the ORAS5 forcing data, leading to improved model performance.
  - Experimental Design: The systematic approach used to assess the sensitivity of the model to different forcing datasets is a strength, as it provides valuable insights into the sources of error and bias.
  - Relevance to Ecosystem Modeling: The paper clearly articulates the relevance of the model for ecosystem modeling, highlighting its potential to drive biogeochemical and ecosystem models aimed at understanding ecological productivity in the Salish Sea.
  - Potential for Future Applications: The authors outline several promising future applications of the model, including biogeochemical modeling, data assimilation, and Lagrangian particle simulations, demonstrating the model's broader utility beyond the scope of this study

- Summary of Weaknesses:

  - Limited Spatial Coverage:

- The model's evaluation is primarily focused on the Strait of Georgia, with limited assessment in other regions, particularly the Puget Sound. This raises concerns about the generalizability of the results to the entire Salish Sea. We have added comments to explain Puget Sound is a priority for evaluation (Line 596) in the future and we point the reader to tables A1-A2.
- Persistent Biases:
  - The model exhibits persistent biases in salinity, especially in the Discovery Islands and Juan de Fuca Strait. The fresh bias in these regions could significantly impact the accuracy of ecosystem models that rely on this data. We made changes to make it clear to the reader that performance is especially compromised in areas with narrow topography such as the Discovery Islands. We also feel we have been transparent about the model biases, quantifying and displaying them in Figures 4-8.
  - While the temperature bias correction is effective, there are still depth dependent biases in temperature, particularly in deeper waters, indicating potential limitations in the model's representation of vertical processes.
- Limited Validation Data:
  - The reliance on a single station (Nanoose) for trend analysis and evaluation of interannual variability limits the robustness of the conclusions. The authors should consider using additional data sources (e.g., satellite data, other stations) to validate the model's performance across a wider range of conditions. Nanoose is the only station where data are available for trend analysis over depths, which we note in the body of the manuscript.
- Simplified Tidal Forcing:
  - The use of only eight tidal constituents and outdated WebTide data might not fully capture the complex tidal dynamics of the Salish Sea, potentially impacting the accuracy of the model's circulation and mixing. See response above.
  - o  Outdated NEMO Version:
  - The use of NEMO v3.6, an older version of the model, might limit the model's capabilities and potential for improvement. Newer versions could offer enhanced numerical schemes and physical parameterizations. We make a note that NEMO 4.x is a priority as a next step (Line 846)
  - Uncertainty Quantification: The paper lacks a rigorous quantification of uncertainties associated with the model's predictions. This makes it difficult to assess the reliability of the results and their implications for future research and applications.

**5. Overacoll Assessment and Recommendation**

- Recommendation:
  - I recommend this paper for publication in GMD after minor revisions. The paper presents a valuable contribution to the field of regional ocean modeling by developing a much-needed hindcast model for the Salish Sea. The model demonstrates good skill in reproducing observed temperature variability and trends, and the authors' systematic approach to evaluating forcing data and bias

correction is commendable. However, the limitations related to spatial coverage, salinity bias, and depth-dependent errors warrant further attention and should be more thoroughly addressed before final publication. We have now added more discussion about the limitations of the model throughout. E.g., salinity bias (Line 836) – though note that the biases are clearly quantified in the tables and figures; spatial coverage and associated weakness of evaluation of Puget Sound is clearly mentioned (830 – 834). With respect to depth dependent errors, we feel our bias correction addresses and improves the depth-dependent errors substantially. We also conducted an evaluation of how time-averaging decreases errors in SST and SSS (e.g., Figure 7) which helps quantify how the errors scale with the chosen time scale.

- Justification:

  o The authors have successfully developed a 38-year hindcast model of the Salish Sea, which is a significant achievement given the limited observational data available for this region. The model's ability to reproduce observed temperature patterns, particularly after bias correction, suggests that it can be a valuable tool for understanding long-term changes and variability in the Salish Sea. The authors' transparency in discussing the limitations of the model, such as its relatively coarse resolution and the lack of data assimilation, is commendable. o However, the model's performance in simulating salinity remains a concern, especially in the Discovery Islands and Juan de Fuca Strait. The depth-dependent biases in temperature also raise questions about the model's ability to accurately represent vertical processes. Additionally, the reliance on a single station for trend analysis and the limited spatial coverage of the evaluation warrant further investigation.

  o To address these limitations, the authors should consider:

    - Expanding the spatial coverage of the evaluation to include Puget Sound and other underrepresented regions.
    - Investigating the sources of the salinity bias and exploring alternative bias correction methods or model refinements to address this issue.
    - Analyzing the model's performance at finer temporal resolutions relevant to ecological processes.
    - Quantifying uncertainties associated with the model's predictions and conducting sensitivity analyses to assess the robustness of the results.
    - Exploring the potential benefits of data assimilation to further improve model accuracy.
    - The authors could also consider moderating their conclusions regarding the model's applicability for ecosystem modeling, given the lack of data assimilation and the remaining biases. While the model shows promise for studying long-term changes, its limitations should be acknowledged and addressed before it can be confidently used for ecological applications.

Overall, this paper is a valuable contribution to the field of regional ocean modeling, but it requires minor revisions to strengthen its conclusions and address its limitations.

Additional comment (from discussion thread):

I realise I also forget in the review issues with the initial condition and flushing time for the spin up. Given it is from a different historic period and the flushing time is of the order 3 years it would seem the 1 year spin up period is in appropriate and may cause issues the authors report in the early years of the hindcast. Thank you, we mention that it is a priority to try a 3 year spin up in future work and provide rationale why we initially went with 1 year. We agree that it may explain the issues early in the hindcast. (Lines 170, 701).

**Citation**: https://doi.org/10.5194/gmd-2024-58-RC2

**6. Detailed Comments:**

- Abstract:

  - Strengths:
    - Concisely summarizes the key points of the paper, including the motivation, methodology, main findings, and potential applications.
    - Clearly states the problem of observational gaps in the Salish Sea and the need for a long-term hindcast model.
    - Highlights the model's skill in reproducing observed temperature trends and anomalies, as well as its potential for supporting ecosystem model development.
  - Weaknesses:
    - Could be more specific about the magnitude and types of biases found in the forcing data and the model.
    - The statement about "new insights" into ocean trends could be more explicit, outlining what specific new information the model provides. Thank you, this has now been revised to be more specific.
    - Could briefly mention the limitations of the model, such as the exclusion of Puget Sound and the persistent bias in salinity.

  Section 1

  - Strengths:
    - Provides a comprehensive overview of the Salish Sea's ecological and economic importance, emphasizing the need for better understanding of its physical oceanography.
    - Clearly articulates the motivation for developing a long-term hindcast model, citing the lack of observational data and the potential impacts of climate change on the ecosystem.
    - Discusses the limitations of existing models for the Salish Sea, justifying the need for a new model like HOTSSea v1.
    - Outlines the specific objectives of the paper, focusing on evaluating the model's performance and investigating potential biases.
  - Weaknesses:

- The discussion of climate change impacts on the Salish Sea could be more specific, mentioning specific examples of observed or projected changes in temperature, salinity, or circulation patterns. Thank you, we provide specific examples in the first introduction paragraph.
- The potential applications of the model for ecosystem modeling could be more clearly articulated, with specific examples of how the model's output could be used to inform ecological research and management. We have revised the introduction to better emphasise specific potential applications in support of ecosystem modeling.

- Section 2. Model Overview:

  o NEMO Version: While the use of NEMO 3.6 is understandable given resource constraints, the authors should acknowledge that newer versions (e.g., NEMO 4.x ) offer potential advantages, such as improved numerical schemes, physical parameterizations, wetting and drying and computational efficiency. A brief discussion of these potential benefits and the rationale for using NEMO 3.6 would strengthen the paper. (done)

  o Model Resolution: The choice of 1.5 km horizontal resolution is a compromise between accuracy and computational cost. However, the authors should discuss the implications of this resolution for capturing fine-scale processes, particularly in areas with complex topography like the Discovery Islands. A sensitivity analysis with different resolutions could be considered in future work. Our experimental model run v0.1 was compared to SalishSeaCast (0.5 km) which serves as a sensitivity analysis to the change in horizontal grid resolution to 1.5 km.

  o Vertical Coordinates: The use of z-level coordinates might not be ideal for the Salish Sea, which is characterized by steep bathymetry and strong stratification. The authors could discuss the potential advantages of using terrain-following (s- or sigma-) coordinates, which might better capture vertical variations in water properties. This is offered in NEMO 4..x and upgrading to 4.x is a next step we now mention.

  o Section 2.2 Boundary Conditions and Forcings:

    - Open Boundary Conditions: The use of a monthly climatology for the northern boundary and outdated WebTide data for tidal forcing could introduce errors in the model. The authors should acknowledge these limitations and discuss potential alternatives, such as using higherresolution reanalysis data or a more recent tidal model. The highest available reanalysis was used, which we note already in the manuscript.

    - River Discharge: The reliance on climatological river discharge data for most rivers (except the Fraser River) could also introduce uncertainties, as it doesn't account for interannual variability. The authors could explore using a hydrological model to provide more realistic and temporally varying freshwater inputs. Yes, this is a potential next step, though no such model exists.

- Section 3. Model Evaluation:

- o Limited Spatial Coverage: The evaluation primarily focuses on the Strait of Georgia, with limited assessment in other regions, especially Puget Sound. This raises concerns about the generalizability of the results. We are not sure that we would expect the trends reported here to generalize to another semi-enclosed sea - what result would be generalizable? The authors should acknowledge this limitation and discuss potential strategies for expanding the evaluation in future work. This is now discussed in the discussion section. (Line 563)
  - o Choice of Metrics:  While the use of CRMSE and WSS is appropriate, the authors could consider additional metrics to provide a more comprehensive evaluation of model performance. For example, they could assess the model's skill in reproducing specific features of the Salish Sea circulation, such as the estuarine exchange flow or the seasonal cycle of stratification. The scope had to be limited in terms of evaluation included in this Model Description paper, though we now note in the discussion that evaluation of circulation is a good idea as a next step.
  - o Statistical Significance: The authors should consistently report the statistical significance of their results, particularly for trend analysis and comparisons between different model versions. This would strengthen the robustness of their conclusions. We do note the statistical significance – in fact the design of Figures 10 and 11 show the CI's indicating statistical significance.
- Section 4. Results and Discussion:

  - o Depth-Dependent Biases: The model exhibits biases in both shallow and deep waters, particularly for temperature. The authors should delve deeper into the potential causes of these biases, such as inaccuracies in the forcing data, limitations in the vertical mixing scheme, or other model parameterizations. Sure, but we reserve this as an area for future work.
  - o Salinity Bias: The persistent fresh bias in salinity, especially in the Discovery Islands and Juan de Fuca Strait, remains a major concern. The authors should investigate the sources of this bias and explore potential model refinements or alternative forcing data to address it. We do, in fact, devote a considerable amount of space in the manuscript to investigation of the source of the bias and we conclude that in arrows with narrow topography the reduced wind strength is most likely responsible and in the Juan de Fuca it is likely a combination of biases inherited from the ocean boundary forcing and from the atmospheric forcings. But, again, we expose the biases and errors such that we lay groundwork and set the direction for future work in this Model Description paper.
  - o Overestimation of Variability: The model tends to overestimate the variability in salinity. This issue warrants further investigation to understand its potential causes and implications for ecosystem modeling.  Future work.
- Section 5. Conclusions:

- Overconfidence: The authors express a high degree of confidence in the model's capabilities despite the identified limitations. They should acknowledge these limitations more explicitly and moderate their conclusions accordingly. We have revised the discussion section to address more explicitly the limitations.

- Future Work: The authors should provide a more detailed roadmap for future model development, including plans to address the identified limitations, explore alternative forcing data and bias correction methods, and expand the model's evaluation to include

other regions and variables. Thank you, we have added a more detailed roadmap for future work now in the discussion section.

---

## Author Comment (AC8)

The authors wish to extend their gratitude to the anonymous reviewer. The manuscript was substantially improved based on their helpful feedback. Please see our responses in red below.

**General feedback:**

SalishSeaCast

The use of SalishSeaCast in your evaluations as opposed to solely relying on limited observations is a great choice. However, I feel that relying so heavily on a model without discussing its limitations and how they could relate to your interpretation of HOTSSea's effectiveness could make your evaluations somewhat misleading. Please include a discussion of SalishSeaCast limitations before presenting comparisons to it. Thank you, we now have added a note explaining that HOTSSea will share limitations of SalishSeaCast's (SSC) due to a lack of wetting and drying capabilities, climatologies used for river runoff, and apparent issues related to vertical mixing (Lines 235 – 240, Line 832). Other aspects of SSC model skill with respect to physical properties have been previously reported (Olson et al., 2020; Soontiens et al., 2016; Soontiens & Allen, 2017), though not necessarily using the same observational data or the subdomain definitions used here; thus our analysis possibly represents a first look at this model's performance in some respects. We feel it is not within the scope of this article to dedicate more space to targeted evaluation of SSC; Figure 4, for example quantifies some pertinent aspects of the SSC model performance.

Python packages

I wouldn't normally reference which python packages are used (ex. lines 113-119), I think that giving access to your code and referencing it at some point (as you have done) is sufficient. It is also difficult to imagine how a package is used without looking at the notebook/script, so I don't think that this information adds much to your paper. Removed the unnecessary specifics, as suggested.

Adding a README to your zenodo and GitHub repository, pointing to which folders one should look into for specific parts of the paper (such as temperature bias correction or SalishSeaCast evaluations), may be more useful to those interested in applying your methods. Readme was present in original submission; however, it could be clearer where the folders are for the analysis and bias correction – it was revised to clarify.

Statistical equations

The equation for widely known statistical metrics do not need to be detailed in the text (equations 1 and 2 for example), just mention which were used to keep the paper concise. It is much more important to describe what the metric reveals about a dataset. Where lesser-known

–

metrics were used, or modifications to traditional ones were made, then it makes sense to keep the equations.   Thank you, this has now been addressed.

Puget Sound Evaluation

It is necessary to draw the line somewhere, and you explain in the outline of future work that the collation of observations in Puget Sound is a necessary next step. However, since a large portion of your evaluations rely on SalishSeaCast, which covers Puget Sound, - it is not clear why Puget Sound was not included in that analysis. I think that the paper would benefit from either the inclusion of this work or a justification for why it was left out. Stressing that hindcasts of Puget Sound have already been done, as you mention on line 81, may do the trick.

Thank you. The original motivation was to synthesize the inputs for the Ecospace ecosystem modeling work focused on the Strait of Georgia and this is stated in the revised manuscript intro. Also, we mention how data was a limiter for evaluation of Puget Sound other models have hindcasted Puget Sound (though not as far back). (Lines 563, 597)

**Line-by-line comments:**

**Line 56 –** Feely et al., (2010) and Ianson et al., (2016) are also worthwhile regional acidification papers to reference here. Thank you – added these (lines 58-59)

**Line 253** - I found this part confusing, explicitly state the difference between v0.14 and v0.16 to increase clarity. Thank you – we made clarifications.

**Table 3 –** I went back to this table many times while reading the text, here are a few suggestions to increase clarity:

- Are HRDPS1 and HRDPS different? If not, keep naming consistent.  – Typo. Fixed.
- I think you are missing the "H" in "RDPS2 (10 km)" – we agree – this was confusing and there was a typo. We fixed it and added a table footnote to clarify.
- I liked the addition of approximate resolution next to HRDPS1 and RDPS in the second row. Include this for all the forcings.  – as suggested, we have added approximate horizontal resolutions for all forcings. We feel it redundant to repeat all horizontal resolutions throughout Table 3, though, when it is the purpose of Table 1 is to do this.
- In your "evaluation purpose" for v0.16 I think that adding that this run went back to the "full" atmospheric forcing resolution would increase clarity on the difference between v0.14 and v0.16.  – Made edits that help increase clarity, as suggested

**Figure 2 –** Add Nanoose station (the back star) to the legend.  - Done

**Table 4 –** Normalize the CTD count in each subdomain by area in order to discuss heterogeneity more accurately in **line 274**.  - Done

–

**Line 287, 293, 297 –** Define acronyms.  - Done

**Lines 302-309 –** This belongs earlier, after line 282 perhaps, where you describe the model depth and time indexing. Tell us (the reader) about what you're conducting statistics on before telling us about the statistical metrics.  – We made the suggested change

**Line 315 –** Accidental paragraph break? – Fixed.

**Line 324 –** I can't find this dataset in Table 2. If I am not mistaken, then why not include it? If you do include it, then I don't believe that the sentence "Canadian buoy data were downloaded…" is needed.  The final row in Table 2 references the buoy data. We agree the sentence is redundant, so we removed it as suggested. We changed 'wave buoys' instrument label in Table 2 to 'buoys' for clarity (since these buoys collect SST and other info than just wave height).

**Line 351 –** "Described above" at the beginning of a new section is confusing, specify which section it was described in.  – we made changes to make it less confusing, referencing the Table summarising experiments explicitly.

**Line 367 –** Remove "it" – Done.

**Line 382 –** I also don't see Nanoose station in table 2, I'm a bit confused about why some observations were not included.  – Nanoose was included in second line of Table 2 but we agree it was buried in the description and was confusing especially because the dataset was prominent to the paper. We added 'Nanoose' to the 'Dataset Title' in Tab. 2 to fix this and modified Fig. 2 to hopefully make it clearer where Nanoose stn is located.

**Line 393 –** Does "depth strata of the closest HOTSSea grid cell" refer to the depth strata described on line 304 or the depth range of the closest grid cell? – Good catch, it was the latter. We have added clarification to address this.

**Lines 434-436 –** Very cool! – Agreed!

**Line 447 –** I'm not familiar with this statistic – how large is "large"? –We made edits to be more specific ('greater than zero' rather than 'large'). The value of the S statistic is constrained only by the number of pairs of data points being compared, so proportional to length of time series.

**Line 494**   Worthwhile to remind the reader of the difference between v0.12 and v0.14 to reveal what this similarity tells you. We made edits to address this.

**Line 497 –** Confusing wording since v0.14 and v0.16 both use ORAS5. We made edits to increase clarity.

–

**Lines 499-505 –** Make this description more clearly comparative between v0.14 and v0.16, at times it is unclear which version you're referring to. Agreed; we did a revision to this paragraph to make it clearer. (Lines 533-)

**Line 508 –** Perhaps I'm mistaken (and if I am than consider rewording) but I read this line as the higher resolution forcing (HRDPS) leading to lower bias than the lower resolution forcing (ERA5). I don't understand why this was unexpected, please expand. Thank you – this has now been reworded to clarify.

**Line 510 –** I interpret "across all depths" to mean here that the bias was lower everywhere. However, later in the paper it seems to mean the average over the whole water column. Make sure not to use this phrase for both applications. Changes made throughout (e.g., Lines 308, 610) for clarity ("across all depths" was referring to the depth stratum 0 – max z).

**Figure 5 –** Impressive changes after bias correction! –Yes, the simple correction made quite a dramatic improvement – it surprised us!

**Line 576 –** nCRMSE is defined with a capital "N" earlier in the paper. Fixed.

**Figure 7a-7b –** Add NCRMSE label to the plots. Done.

**Figure 7c-7d –** NCRMSE instead of NCRMSD; Done.

**Lines 615-618 –** These sentences seem to contradict. Is SSS variability overestimated or is it good? Thank you, we revised this paragraph to address this.

**R2-M26 - Line 623 –** In addition to a discussion of the time resolution required for HOTSSea to support ecosystem modelling this paper needs a discussion of what spatial resolution is required. Explain to the reader why 1.5 km is good enough. Thank you, more on this was definitely merited. Justification is now given in paragraph 2 in the intro, paragraph 2 in discussion (Line 790), and Line 149.

**Line 630 –** PSU? Yes - fixed

**Line 630 –** remove "be" Done

**Line 648 –** This seems like a rather large salinity bias at the surface to me. Put it into context. What is it in SalishSeaCast? Do you think that this surface salinity bias will affect model circulation? We expanded on this. The bias is shared with SalishSeaCast, as was apparent in Figure 4 in the SGS subdomain. Yes, it is most likely affecting model circulation. (Line 674, 831-835). We prioritise upgrading our model from NEMO 3.6 to 4.x before further exploration of

–

bias correction, since the newer version offers additional features with respect to vertical and lateral mixing.

**Line 689 –** How does it correlate? Negatively. Clarification made. (Line 717)

**Figure 11 –** Could there be a better way to express in the caption that it is a seasonal average of decadal trends? I think solely calling it "seasonal" in the caption could be a bit confusing. Agreed. Change made for clarity.

**Line 750**       Is it your intension to conduct a similar bias correction on salinity despite the instability problems you mentioned on line 369? Yes, we added a note related to this (Line 822,

**Line 771 –** Missing space between Figure 9 and Figure 10 Fixed

**Table A1 –** Something cutoff under "Model" in the top row Fixed

**Figure A3 –** Just a note that I think that the description of these Taylor diagrams is more clear than what you have in the main body. Thanks, we tried to clarify more in Fig 7 caption.

Thank you again for the helpful review. Please note, we are not sure the line numbers (referencing the revised manuscript) we used here will be that helpful since the final manuscript will not include them. However, we are instructed not to re-submit the manuscript in this discussion section. We would be happy to provide the revised Word file upon request if it helps, though.

---

## Referee Report (RR1)

**Feedback**

I commend the authors in the amount of work they put into this new draft. My overarching feedback at this stage is that a manuscript should aim to be "easy" for a reader - state explicitly what you mean, explain what the takeaways from each figure should be, outline exactly how you came to major decisions, and important results or background may be worth repeating. All those areas have been improved upon significantly in this draft, my feedback focuses on a couple areas that I still think could be made easier. I think that with these two relatively small edits that the paper will be ready for publication.

**Evaluations with SalishSeaCast**

Some limitations of SalishSeaCast that may impact HOTSSea have been added, but focus on model design choices or NEMO idiosyncrasies, not on how those impact overall model accuracy. Your discussion of the salinity bias observed in figure 4 (line 675) is effective in outlining that a significant portion of the salinity bias is inherited as well (I would suggest referencing figure 4 again somewhere in lines 831-835, because that is where the inheritance is shown most clearly). I understand your hesitancy to perform or outline specific evaluations of SalishSeaCast, you are right in saying that it is not within the scope of this paper; however, stating explicitly in the methods section the specific (previously reported) aspects of SalishSeaCast that may impact the accuracy of HOTSSea would add credibility to your evaluations. Overall, this concern has been much improved upon.

**Puget Sound**

In the introduction you discuss the ecosystem model focused on the Strait of Georgia being the motivation for HOTSSea, but in the same paragraph you explain that the spatial domain chosen to work effectively with this ecosystem model encompasses all the Salish Sea (including Puget Sound). As such, I don't think that this section in the introduction sufficiently explains to the reader why Puget Sound was left out of your analysis. As it currently stands, the reader would need to already have a keen understanding of Salish Sea circulation and to take the same logical steps that you did without explanation (ie. since the Strait of Georgia is the focus you need to get Juan de Fuca and the Gulf Islands right but Puget Sound is probably less important).

Earlier, on line 88-91 you state "It is therefore one aim of this study to develop a physical hindcast with a spatial-temporal resolution that enables a long hindcast while maintaining acceptable model skill for supporting marine ecological research and ecosystem management." This line comes directly after you explain that previous hindcasts have been

performed for Puget Sound. I think that said line might be an appropriate location to express that performing an accurate hindcast within Puget Sound was deemed less important than for the other regions.

As there is quite a bit of data available in Puget Sound, just not collated for the purpose of this project, and since a comparison to SalishSeaCast results within Puget Sound would have been possible, I don't think that data availability on its own is a reasonable explanation for the lack of Puget Sound evaluation. State explicitly how you decided that while Puget Sound had to be included in the model domain, that an evaluation of it was a lower priority and outside of the scope of this paper.

**Smaller comments**

- "Hare" should be "Haro" in table 4
- Your discussion of the differences between v0.12, v0.14, and v0.16 has been made much clearer.
- Discussion of the spatial scales required to study salmon strengthen the motivation of the paper!

---

## Author Response (AR3)

The authors wish to extend their gratitude to the reviewer for their helpful feedback which we feel has substantially improved the manuscript.

**Detailed list of revisions:**

**Abstract**

- Lines 15 – 10: minor grammatical edits to be more concise.
- Line 20: Defined NEMO acronym before using it.
- Lines 20 – 34: minor grammatical edits to be more clear and concise.
- Lines 33-34: concluding, forward-looking sentence

**Non-Technical Summary**

- Lines 37-44 - revised for clarity, to make it easier for reader to understand overarching aim and findings of the paper.

**1 – Introduction**

- Lines 63, 67, 77, 80, 105 – minor grammar edits.
- Lines 91- 94 ("It is therefore one aim...") - added clarity as to why we did not focus evaluation on Puget Sound, as the reviewer suggested.
- Lines 97,103-104 – grammar edits for clarity

**2.1- Spatial-Temporal Configuration**

- Lines 140-143 – edits for clarity, grammar
- Line 144 ("To ensure dynamics...") – edit to address reviewer's request to improve clarity that a main motivator is resolving the central and northern portion (Strait of Georgia). Emphasis on the importance of Juan de Fuca and Southern Gulf for resolving circulation, which we feel implies that PS and DI are not quite as important and hence why evaluation efforts were focused of JFS,SGS, SGN, SGI and not as much on PS.
- Lines 154,157, 173 – minor grammar edits

**3. - Methods**

- Lines 242 – 252 ("As such, HOTSSea v1 shares...") - In the first paragraph, we previously mentioned that we used SalishSeaCast to compare with HOTSSea results. To address the reviewer's suggestion to "focus on model design choices or NEMO idiosyncrasies" we explicitly mention several of these idiosyncrasies, followed by specifics of the previously reported performance of SSC.

**3.1 - Methods**

- Line 266 – (Near top of the first paragraph) - we previously mentioned that SalishSeaCast has already been evaluated. In response to the reviewer's request,

we add some specifics about the model and we moved this to Section 3 to put it more front and center (see above).

**3.2.1 – Vertical Profiles**

- Line 312 (Table 4) – typo corrected.

**4.4.2. – Strait of Georgia**

- Line 755 – minor edit to split sentences

**5 – Discussion and Conclusion**

- Lines 774-779,780,785-788, 792 (first paragraph) – minor edits for flow, changes to grammar.
- Lines 796, 802 (second paragraph) – minor edits to be more concise.
- Lines 810-815 (third paragraph) – re-ordered sentences to emphasis a key contribution of the paper, trimmed some sentences, tried to be more concise
- Lines 825, 830-835 (third paragraph) – minor grammar edits
- Line 838 (fourth paragraph) – added reference to supplemental tables S1,S2 which have PS evaluation metrics
- Lines 843 (fourth paragraph) - added another reference to Figure 4 (as reviewer suggested).
- Line 847 (fourth paragraph)  – minor grammar edit
- Lines 857-863 (fifth paragraph) – added reference to Fig S4, moved references to next sentence, minor edits to be more concise, clear
- Line 878, 881 (final paragraph) – grammar edit to be more concise

**Code and Data Availability**

- Lines 916 – 920 – Added explicit reference to Zenodo having the monthly processed outputs to be consistent. Updated line referring to 'pacea' project.

**References**

- Line 1016 - Addition of URL link to github of pacea repo
- Line 1214- Update to Zenodo version of data outputs (Oldford et al (a)).

**Supplemental**

- Tables S1 and S2 – padding and spacing fixes so they each fit on one page
- Tables S1 and S2 – changed from 'A1' and 'A2' to align with GMD guidelines

**Other:**

- Figure and Table reference style made consistent  (e.g., 'Tables' instead of 'Tabs.')
- Abbreviation 'Fig.' and 'Tab.' used when in sentence (GMD guideline)

**Feedback**

I commend the authors in the amount of work they put into this new draft. My overarching feedback at this stage is that a manuscript should aim to be "easy" for a reader - state explicitly what you mean, explain what the takeaways from each figure should be, outline exactly how you came to major decisions, and important results or background may be worth repeating. All those areas have been improved upon significantly in this draft, my feedback focuses on a couple areas that I still think could be made easier. I think that with these two relatively small edits that the paper will be ready for publication.

Thank you for the feedback. We agree.

We made edits to make it clearer and more concise – overall, easier for the reader (e.g., Lines 15-35, 91-94,140-147, 809-815).

**Evaluations with SalishSeaCast**

Some limitations of SalishSeaCast that may impact HOTSSea have been added, but focus on model design choices or NEMO idiosyncrasies, not on how those impact overall model accuracy. Your discussion of the salinity bias observed in figure 4 (line 675) is effective in outlining that a significant portion of the salinity bias is inherited as well (I would suggest referencing figure 4 again somewhere in lines 831-835, because that is where the inheritance is shown most clearly).

Thank you. Changes were made to address this (Lines 857-861).

I understand your hesitancy to perform or outline specific evaluations of SalishSeaCast, you are right in saying that it is not within the scope of this paper; however, stating explicitly in the methods section the specific (previously reported) aspects of SalishSeaCast that may impact the accuracy of HOTSSea would add credibility to your evaluations. Overall, this concern has been much improved upon.

Thank you. We have now revised to be more explicit, clear about the model idiosyncrasies and design choices associated with NEMO and shared with the particular the SSC model design - in Section 3 (Lines 242-256).

**Puget Sound**

In the introduction you discuss the ecosystem model focused on the Strait of Georgia being the motivation for HOTSSea, but in the same paragraph you explain that the spatial domain chosen to work effectively with this ecosystem model encompasses all the Salish Sea (including Puget Sound). As such, I don't think that this section in the introduction sufficiently explains to the reader why Puget Sound was left out of your analysis.

We think you are referring to Section 2.1, not the introduction (Section 1), since this is where the motivation and domain is described. Actually, PS was not entirely omitted from our analysis – we just did not have much data collated and wrangled. The results are reported in the Supplemental tables (S1-S2). We added two more references to these tables for the reader (Lines 843, 857-862)

As it currently stands, the reader would need to already have a keen understanding of Salish Sea circulation and to take the same logical steps that you did without explanation (ie. since the Strait of Georgia is the focus you need to get Juan de Fuca and the Gulf Islands right but Puget Sound is probably less important).

Earlier, on line 88-91 you state "It is therefore one aim of this study to develop a physical hindcast with a spatial-temporal resolution that enables a long hindcast while maintaining acceptable model skill for supporting marine ecological research and ecosystem management." This line comes directly after you explain that previous hindcasts have been performed for Puget Sound. I think that said line might be an appropriate location to express that performing an accurate hindcast within Puget Sound was deemed less important than for the other regions. As there is quite a bit of data available in Puget Sound, just not collated for the purpose of this project, and since a comparison to SalishSeaCast results within Puget Sound would have been possible, I don't think that data availability on its own is a reasonable explanation for the lack of Puget Sound evaluation. State explicitly how you decided that while Puget Sound had to be included in the model domain, that an evaluation of it was a lower priority and outside of the scope of this paper.

Thank you, yes. You have the 'logical steps' exactly correct and we agree it was not articulated well enough. We have edited the sentence formerly on line 88 – 91, as suggested (Lines 91-95). Changes at Lines 144 also aim to clarify.

**Smaller comments**
• "Hare" should be "Haro" in table 4 – fixed (Line 312).
• Your discussion of the differences between v0.12, v0.14, and v0.16 has been made much clearer. – thanks!
• Discussion of the spatial scales required to study salmon strengthen the motivation of the paper! – thanks!